# The effect of variation of individual infectiousness on SARS-CoV-2 transmission in households

Tim K Tsang[1,2]*, Xiaotong Huang[1], Can Wang[1], Sijie Chen[1], Bingyi Yang[1], Simon Cauchemez[3†], Benjamin John Cowling[2]*†

[1]WHO Collaborating Centre for Infectious Disease Epidemiology and Control, School of Public Health, Li Ka Shing Faculty of Medicine, The University of Hong Kong, Hong Kong, China; [2]Laboratory of Data Discovery for Health, Hong Kong, China; [3]Mathematical Modelling of Infectious Diseases Unit, Institut Pasteur, Paris, France

**Abstract** Quantifying variation of individual infectiousness is critical to inform disease control. Previous studies reported substantial heterogeneity in transmission of many infectious diseases including SARS-CoV-2. However, those results are difficult to interpret since the number of contacts is rarely considered in such approaches. Here, we analyze data from 17 SARS-CoV-2 household transmission studies conducted in periods dominated by ancestral strains, in which the number of contacts was known. By fitting individual-based household transmission models to these data, accounting for number of contacts and baseline transmission probabilities, the pooled estimate suggests that the 20% most infectious cases have 3.1-fold (95% confidence interval: 2.2- to 4.2-fold) higher infectiousness than average cases, which is consistent with the observed heterogeneity in viral shedding. Household data can inform the estimation of transmission heterogeneity, which is important for epidemic management.

**\*For correspondence:**
timtsang@connect.hku.hk (TKT);
bcowling@hku.hk (BJC)

†These authors contributed equally to this work

## Editor's evaluation

While it has been demonstrated that for SARS-CoV-2, a small fraction of individuals contributes to the majority of onward transmission, this heterogeneity is driven by multiple factors that span both biological and behavioral causes. By performing a solid meta-analysis of household transmission studies, the authors fit a household transmission model to the curated data to estimate variation in infectiousness which provides a valuable contribution to the existing knowledge base. By collating data from multiple studies, they are able to more fully investigate individual variability.

## Introduction

Characterizing transmission is critical to control the spread of an emerging infectious disease. The reproductive number is the widely adopted measure of infectiousness. However, it only measures the average number of secondary cases infected by an infected person, not the heterogeneity in the number of transmissions. Variation of individual infectiousness is particularly highlighted by superspreading events (SSEs), in which a minority of cases are responsible for a majority of transmission events. Such phenomena, illustrated by the '80/20 rule' (i.e., 20% of cases responsible for 80% transmission; *Adam et al., 2020*; *Sun et al., 2021*), have been observed in emerging infectious disease outbreaks (*Lloyd-Smith et al., 2005*), including severe acute respiratory syndrome (SARS) (*Shen et al., 2004*), Middle East respiratory syndrome (*Cauchemez et al., 2016*; *Cowling et al., 2015*), and most recently the COVID-19 pandemic (*Adam et al., 2020*; *Lau et al., 2020*; *Sun et al., 2021*;

*Wong and Collins, 2020*; *Zhao et al., 2021*). In these outbreaks, the proportion of cases attributed to 80% transmission, and the dispersion parameter that is estimated by fitting the negative binomial distribution to the number of secondary cases (*Lloyd-Smith et al., 2005*), are considered as measures of transmission heterogeneity.

However, the number of contacts per index cases is often not reported in SSE studies, and hence not incorporated in the analyses. In addition, SSE studies usually analyze clusters from different settings, in which the baseline transmission risk and density of exposure could be different (*Thompson, 2021*). Finally, studies of transmission heterogeneity that focus on SSEs described in the literature may suffer from publication bias, with larger clusters having higher probability of being observed and reported (*Zhao et al., 2021*). Therefore, the observed heterogeneity in the number of secondary cases could be the result of large number of contacts in SSE settings, or confounding from these factors (*Althouse et al., 2020*; *Bagdasarian and Fisher, 2020*), instead of variation in individual infectiousness.

Households are one of the most important settings for SARS-CoV-2 transmission, with 4- to 10-fold higher transmission risk than other places (*Thompson, 2021*). Hence, household transmission studies provide an ideal setting to quantify variations in individual infectiousness. In a household transmission study, an index case is identified, and their household contacts are followed up for 1–2 weeks, during which there is high transmission potential (*Tsang et al., 2016*). Therefore, the number of contacts is known while transmission risks and reporting biases can be controlled. We aim to characterize the variation of individual infectiousness by analyzing data from household transmission studies.

## Results

We conducted a systematic review to gather information on the number of secondary cases with the number of household contacts for each household, in the form of number of households with *X* cases among households of size *Y*. In total, we identified 17 studies, comprising 13,098 index cases and 31,359 household contacts (*Appendix 1—figure 1*, *Appendix 1—table 1*; *Bernardes-Souza et al., 2021*; *Carazo et al., 2022*; *Dattner et al., 2021*; *Han et al., 2022*; *Hart et al., 2022*; *Hsu et al., 2021*; *Hubiche et al., 2021*; *Koureas et al., 2021*; *Laxminarayan et al., 2020*; *Layan et al., 2022*; *Lyngse et al., 2022*; *Méndez-Echevarría et al., 2021*; *Posfay-Barbe et al., 2020*; *Reukers et al., 2021*; *Wilkinson et al., 2021*). Most studies covered a period from January to November 2020, which was dominated by ancestral strains, except for Layan et al. (from December 2020 to April 2021) and Hsu et al. (from January 2020 to February 2021), which covered both ancestral strains and the alpha variant.

We then developed a statistical model to quantify the degree and the impact of variation of infectiousness of cases on transmission dynamics. The individual-based household transmission model describes the probability of infection of household contacts as depending on the time since infection in other infected persons in the household, so that infections from outside the household (community

**Figure 1.** Summary of statistics for 17 identified studies. Figure shows the average number of contacts and standard deviation (SD) of number of contact, SD of number of secondary cases per index cases ($\sigma_{sec}$), and secondary attack rate (SAR) for 17 identified studies.

| Article | Infectiousness variation ($\sigma_{var}$) | Probability of infection from community ($10^{-2}$) | Probability of infection from household | ΔDIC |
|---|---|---|---|---|
| Lyngse, et al. | 1.48 (1.29, 1.7) | 0.06 (0, 0.16) | 0.1 (0.08, 0.12) | 230.1 |
| Carazo, et al. | 1.41 (1.19, 1.72) | 0.2 (0.01, 0.43) | 0.3 (0.24, 0.34) | 253.5 |
| Laxminarayan, et al. | 2.44 (1.98, 3.23) | 0.03 (0, 0.11) | 0.04 (0.01, 0.07) | 268.7 |
| Dattner, et al. | 1.12 (0.65, 1.69) | 0.63 (0.12, 1.06) | 0.31 (0.19, 0.42) | 29.2 |
| Layan, et al. | 1.12 (0.74, 1.76) | 0.3 (0.02, 0.96) | 0.26 (0.15, 0.39) | 15.6 |
| Hart, et al. | 0.35 (0.11, 0.96) | 0.74 (0.05, 2.12) | 0.51 (0.32, 0.65) | 3.8 |
| Hubiche, et al. | 1.03 (0.45, 2.68) | 1.24 (0.14, 2.48) | 0.32 (0.06, 0.57) | 5.8 |
| Wilkinson, et al. | 1.05 (0.12, 3.51) | 0.32 (0.02, 0.91) | 0.07 (0, 0.18) | 6.1 |
| Tsang, et al. | 2.83 (1.48, 4.73) | 0.45 (0.05, 1.03) | 0.06 (0, 0.26) | 25.9 |
| Reukers, et al. | 1.79 (0.72, 4.5) | 1.53 (0.18, 2.93) | 0.22 (0.01, 0.6) | 9.4 |
| Han, et al. | 1.71 (0.51, 4.13) | 1.67 (0.19, 2.99) | 0.21 (0.02, 0.61) | 7.4 |
| Méndez–Echevarría, et al. | 0.51 (0.13, 1.7) | 0.91 (0.05, 2.53) | 0.28 (0.06, 0.52) | 1.5 |
| Dutta, et al. | 1.53 (0.5, 4.2) | 1.1 (0.12, 2.26) | 0.24 (0.01, 0.63) | 11 |
| Koureas, et al. | 1.88 (0.6, 3.66) | 0.84 (0.1, 1.76) | 0.13 (0.01, 0.42) | 24.7 |

**Figure 2.** Modeling results of household transmission dynamics and infectiousness variation. Figure shows the estimates of infectiousness variation ($\sigma_{var}$), the estimated probability of infection from community and estimated probability of infection from households, and the reduction in deviance information criteriion (DIC) compared with the model without infectiousness variation. Models are fitted separately to 14 household transmission studies.

infections), or infections via other household contacts rather than the index case (tertiary infections) are allowed (*Cauchemez et al., 2009*; *Tsang et al., 2014*; *Tsang et al., 2021*). Therefore, the model could estimate the per-contact hazard of infection, which implicitly controls for number of household contacts in households. We extend this model by adding a random effect ($\delta_i$) on the individual infectiousness of cases. Here, the relative infectiousness of case $i$ compared with case $j$ is $\exp(\delta_i)/\exp(\delta_j)$. The parameter for variation in individual infectiousness (hereafter denoted as infectiousness variation, $\sigma_{var}$) is the standard deviation (SD) of the random effect characterizing individual infectiousness, so that $\delta_i$ follows a normal distribution with mean equal to 0 and SD equal to $\sigma_{var}$.

We separately fit the models to 14 studies with more than 150 contacts (*Carazo et al., 2022*; *Dattner et al., 2021*; *Dutta et al., 2020*; *Han et al., 2022*; *Hart et al., 2022*; *Hubiche et al., 2021*; *Koureas et al., 2021*; *Laxminarayan et al., 2020*; *Layan et al., 2022*; *Lyngse et al., 2022*; *Méndez-Echevarría et al., 2021*; *Reukers et al., 2021*; *Tsang et al., 2022*; *Wilkinson et al., 2021*; *Figure 1*, *Appendix 1—table 2*). For 12 studies out of 14, models with infectiousness variation perform substantially better (range of ΔDIC: 5.8–268) (*Figure 2*, *Appendix 1—table 2*). From these 12 studies, the estimated infectiousness variation ($\sigma_{var}$ ranged from 1.03 to 2.83. This suggests that, the 20% most infectious cases are 2.4- to 10-fold more infectious than the average case. Based on the two largest studies with 6782 and 3727 households (*Carazo et al., 2022*; *Lyngse et al., 2022*), the estimated infectiousness variation is 1.48 (95% credible interval [CrI]: 1.29, 1.7) and 1.41 (95% CrI: 1.19, 1.72), suggesting that, among all cases, the 20% most infectious are 3.5-fold (95% CrI: 3.0- to 4.2-fold) and 3.3-fold (95% CrI: 2.7- to 4.3-fold) more infectious than the average case. The estimated daily probability of infection from outside the household and estimated person-to-person transmission probability within households range from 0.0003 to 0.017, and from 0.06 to 0.51, respectively. The estimates of parameters for the relationship between number of contacts and transmission (larger value indicates stronger inverse association) range from 0.43 to 0.92, except for the study by *Layan et al., 2022*, where it is equal to 0.2. For all studies, the predicted final size distribution is consistent with the observed data and the model fit is judged adequate (*Appendix 1—Tables 3–7*, *Appendix 1—figure 2*). Simulation studies demonstrated that there is no important systematic bias, with 88–100% (depending on the parameter)

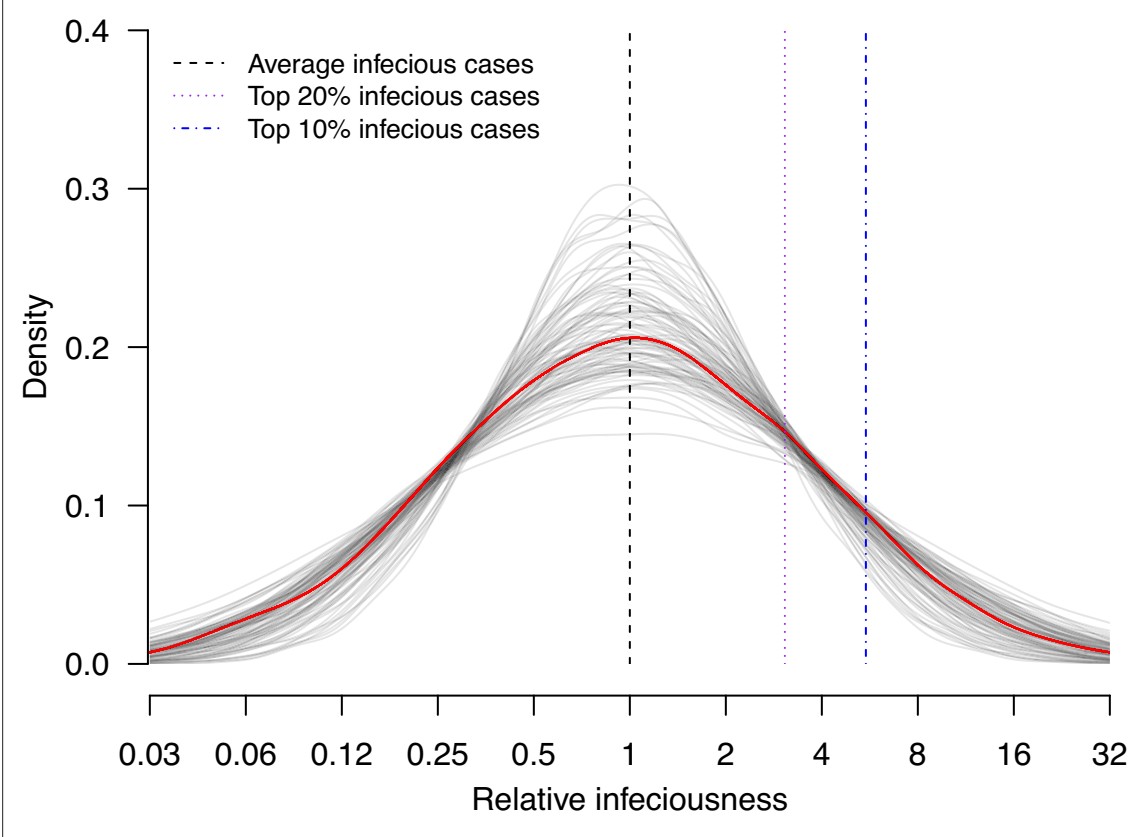

**Figure 3.** Estimate distribution of relative infectiousness based on the pooled estimate. Red line indicates the estimated distribution and the gray lines indicate the associated uncertainty. Black dashed line indicates average infectiousness (relative infectiousness equal to 1), while the purple and blue dashed lines indicate top 20% and 10% infectiousness, respectively.

of the 95% credible intervals covering the simulation value (*Appendix 1—table 8*). This suggests that the algorithm could estimate adequately the posterior distribution. We conduct a sensitivity analysis that assumes the individual infectiousness of cases follows a Gamma distribution, but the fit worsens substantially (*Appendix 1—table 9*).

We conduct random effects meta-analyses on estimates of individual infectiousness from the 14 identified studies. The pooled estimate of infectiousness variation is 1.33 (95% confidence interval [CI]: 0.95, 1.70), suggesting that the 20% most infectious cases are 3.1-fold (95% CI: 2.2- to 4.2-fold) more infectious than the average case (*Figure 3*). Based on this fitted distribution, we estimate that 5.9% (95% CI: 1.4%, 11.1%) and 14.9% (95% CI: 7.2%, 20.7%) of cases could be at least 8- and 4-fold more infectious than average cases, respectively.

We further explore if the secondary attack rate (SAR, the proportion of infected contacts), and the SD of the distribution of number of secondary cases ($\sigma_{sec}$ (*Figure 4*; *Appendix 1—table 10*) may be correlated with the infectiousness variation. In meta-regression, we find the infectiousness variation is associated with SAR (*Appendix 1—table 11*). We estimate that doubling SAR are associated with 0.55 (95% CI: 0.21, 0.89) unit decrease in infectiousness variation, with R-squared equal to 67% respectively. In addition, we find that higher infectiousness variation is associated with only using PCR to ascertain secondary cases. Regarding other statistics, we find that $\sigma_{sec}$ is positively associated with mean and SD of number of contacts. Other than these associations, we find no association between these statistics and implementation of lockdown, ascertainment method of index and secondary cases, and the circulating virus of SARS-CoV-2 in the study period.

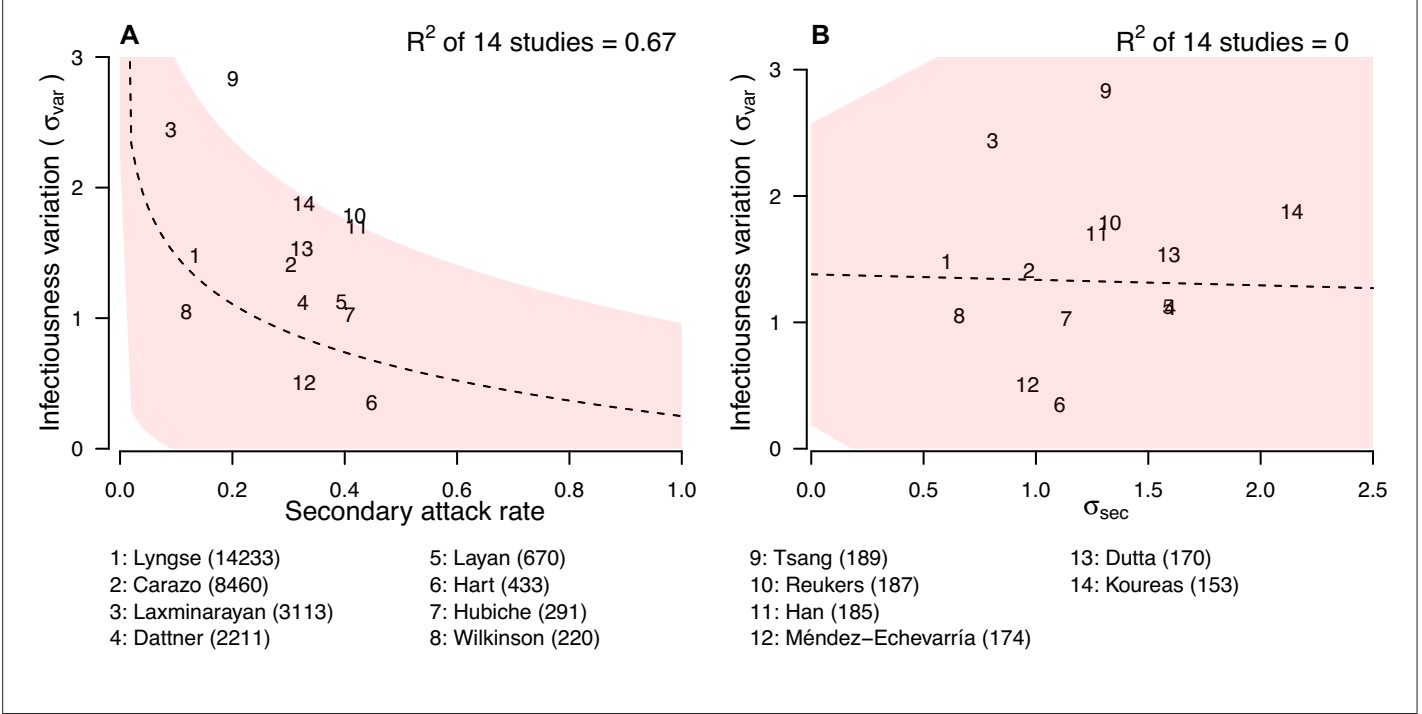

**Figure 4.** Relationship between infectiousness variation and statistic. In each panel, numbers represent the observed corresponding relationship for the identified studies. Panels A and B show the relationship between infectiousness variation ($\sigma_{var}$) and secondary attack rate (SAR) and standard deviation (SD) of number of secondary cases per index cases ($\sigma_{sec}$). In the bottom, numbers in bracket indicate the number of household contacts in corresponding studies.

## Discussion

In this study, we characterize the impact of variation of individual infectiousness on heterogeneity of transmission of COVID-19 in households. We demonstrate that it can be estimated from household data using a modeling approach. The pooled estimate of infectiousness variation from 14 studies suggests that the 20% most infectious cases have 3.1-fold (95% CI: 2.2- to 4.2-fold) higher infectiousness compared with average index cases. This implies there is substantial variation in individual infectiousness of cases in households. Given that we focused our analysis on households with known number of contacts in studies conducted in the early outbreaks caused by ancestral strains, the estimated infectiousness variations are corrected for the variations caused by number of contacts and transmission risks in different settings, difference in pre-existing immunity among contacts (almost everyone is naïve and unvaccinated), and the difference in transmissibility among variants. Hence, this estimated infectiousness variation measures the variation caused by difference in individuals, which may be contributed by both biological factors and host behaviors, but not other potential confounding factors mentioned above.

Regarding host behaviors, multiple contact patterns, particularly by age, could contribute to the variations in infectiousness of cases. For example, mother-child contacts are usually more intense than father-child contacts, and adult cases are more capable of self-isolation within a household compared with children. Furthermore, contact pattern studies suggested that school-age children and young adults tended to mix with people of the same age (*Mossong et al., 2008*; *Mousa et al., 2021*). Also, the duration of contact could vary, which could also contribute to the variations in infectiousness of cases (*Toth et al., 2015*). It should be noted that contacts have different features, including their number, frequency, and duration that may contribute to variations in infectiousness. Previous studies also suggest that the relative importance of contact characteristics on transmission may be different among viruses (*De Cao et al., 2014*). Individual behaviors may also be influenced by control measures and recommendation from public health agencies. For example, lockdown and stay-at-home orders may increase the time spending at home. Mask-wearing when in contact with other household members, using separate bedrooms and bathrooms, avoiding having meals together may

reduce the transmission in households (**Ng et al., 2021**). Also, social disparity such as occupation may increase or reduce the risk of transmission in households, including the availability of personal protective equipment (PPE), or being healthcare worker (**Yang et al., 2021**). These factors may have impact of the transmission risk and hence SAR in households. In addition, heterogeneity in these different factors may have contributed infectiousness variation.

Biological factors may also contribute to such variations. For example, the SAR for index cases with fever and cough are 1.4- and 1.3-fold higher than index cases without fever and cough, respectively (**Madewell et al., 2021**). Viral shedding is used as a proxy measure of infectiousness, and consistently it also has substantial variations, as suggested by 11 systematic reviews (**Appendix 1—Tables 12–14**). Regarding the magnitude of viral shedding, studies report high variations of temporal viral shedding patterns among individuals (**He et al., 2020**; **Jones et al., 2021**; **Sun et al., 2022**). In addition, the duration of viral shedding can be highly heterogeneous, with pooled estimates of mean durations ranging from 11.1 to 30.3 days, and almost all reviews reporting high heterogeneity of estimates (**Cevik et al., 2021**; **Chen et al., 2021b**; **Díaz et al., 2022**; **Fontana et al., 2021**; **Li et al., 2020b**; **Okita et al., 2022**; **Qutub et al., 2022**; **Rahmani et al., 2022**; **Xu et al., 2020**; **Yan et al., 2021**; **Zhang et al., 2021**). Such heterogeneities still exist in subgroup analyses by age and severity (**Cevik et al., 2021**; **Fontana et al., 2021**; **Okita et al., 2022**; **Qutub et al., 2022**; **Rahmani et al., 2022**; **Xu et al., 2020**; **Yan et al., 2021**). The infectious period, proxied by duration of replicant competent virus isolation, is also heterogeneous (**Rahmani et al., 2022**; **Appendix 1—table 14**). One review also suggests that heterogeneity in viral shedding is an intrinsic virological factor facilitating higher dispersion parameter for SARS-CoV-2 if we compare it with the corresponding patterns in SARS-CoV-1 and pandemic influenza A(H1N1) pdm09 (**Chen et al., 2021a**).

The observed variation in individual infectiousness is consistent with past analyses of the dispersion parameter in a negative binomial distribution fitted to number of secondary cases per index case of COVID-19 (**Sun et al., 2021**; **Wong et al., 2015**). However, in these studies, the number of contacts was not considered. Therefore, the observed variations may not apply directly to households with limited number of contacts. It should be noted that the SD of the distribution of number of secondary cases ($\sigma_{sec}$) is only weakly correlated with infectiousness variation. This is because $\sigma_{sec}$ is highly correlated with the mean and SD of the number of contacts, suggesting that it may depend on distribution of number of contacts, and hence may not be comparable among studies. Infectiousness variation is also correlated with the SAR. When SAR is higher, it is expected that more contacts are infected and therefore the observed number of secondary cases is less heterogeneous (**Adam et al., 2022**). We find higher infectiousness variation is associated with only using PCR to confirm secondary cases. One potential reason is that using other methods may lower the sensitivity of detecting infection, resulting in lower estimates of SAR and hence higher estimates of infectiousness variation. Further investigations are needed to explore roles of other potential factors affecting infectiousness variation, such as contact frequency among different regions (**Mousa et al., 2021**).

An important limitation of our study is that we do not have individual-level data. Therefore, we are unable to determine the impact of demographic factors like age and sex on infectiousness variation. Also, we cannot disentangle the host behaviors from biological factors. Previous analysis suggested no evidence of the impact of age on infectiousness of cases (**Lau et al., 2020**; **Sun et al., 2021**). In addition, we could not include factors affecting susceptibility to infection. Our estimates of infectiousness variation should be interpreted in light of these limitations: they capture heterogeneity in infectiousness due to demographic, host, and biological factors. However, in one study that included susceptibility component in the estimation of individual infectiousness, substantial heterogeneity remained with 20% of cases estimated to contribute to 80% of transmission (**Tsang et al., 2022**). Second, the recruitment methods among studies are different. This may affect the comparability of the results, although all index cases are laboratory-confirmed in all studies (**Tsang et al., 2021**). Third, we assumed that risks of infection from community for all households are the same, but there were different factors that may affecting this, including occupations, such as healthcare workers, social economic status that related to assess to PPEs (**Yang et al., 2021**). Finally, most of our identified studies were conducted in the period of circulation of ancestral strains, and therefore the identified infectiousness variation may not be directly applicable to other variants.

In conclusion, we developed a modeling approach to estimate variation in individual infectiousness from household data. Result indicates that there is substantial variation in individual infectiousness, which is important for epidemic management.

## Materials and methods
### Study design

The aim of this study was to develop a statistical model to quantify the variation of individual infectiousness in households, based on publicly available information. An index case was defined as the first detected case in a household, while secondary cases were defined as the identified infected household contacts of the index case. We conducted a systematic review to collect household studies with at least 30 households, reporting the number of secondary cases with number of household contacts for each household for COVID-19, in the form of number of households with $X$ cases among households of size $Y$. For each study, we also extracted the study period, the coverage of tests of household contacts, the case ascertainment methods, the circulating virus of SARS-CoV-2, and the public health and social measures in the study period. This information was used as an input of modeling analyses in this study. Details of systematic review could be found in Appendix 1.

### Estimation of variation of individual infectiousness in households

To determine if there are variations of individual infectiousness of cases, we used an individual-based household transmission model (; *Tsang et al., 2014*; *Tsang et al., 2021*). The model described the probability of infection of household contacts as depending on the time since infection in other infected people in the household, while infections from outside the household (community infections), or infections via other household contacts rather than the index case (tertiary infections) are allowed. We extended the model by adding a random effect parameter ($\delta_i$) on the individual infectiousness of each case. In the model, the hazard of infection of individual $j$ at time $t$ from an infected household member $i$, with infection time $t_i$ in household $k$, was

$$\lambda_{i \to j}\left(t\right) = \frac{\lambda_h}{X_k^{\beta}} * \exp\left(\delta_i\right) * f\left(t - t_i\right)$$

where $\lambda_h$ was the baseline hazard, $\delta_i$ followed a normal distribution with mean 0 and SD $\sigma_{var}$, which quantified the variation of individual infectiousness (hereafter denoted as infectiousness variation). The relative infectiousness of case $i$ compared with case $j$ was $\exp(\delta_i)/\exp(\delta_j)$.

$X_k$ was the number of household contacts. $\beta$ was the parameter describing the relationship between number of household contacts and transmission rate. It ranged from 0 to 1, with 0 indicating that the transmission rate was independent of number of household contacts while 1 indicated that the transmission rate was inversely proportional to number of household contacts (i.e. dilution effect of the contact time per contact which was lower when the number of household contacts is higher). $f(\cdot)$ was the infectiousness profile since infection generated from the assumed incubation period (mean equal to 5 days) and infectious period (mean equal to 13) (*He et al., 2020*; *Jing et al., 2020*; *Li et al., 2020a*; *Appendix 1—table 15*). Extracted distribution were summarized in *Appendix 1—table 15*, and the shape of the assumed function was plotted on *Appendix 1—figure 3*.

Since the data were extracted from publication, the infection time of all cases was unavailable for all studies. Also, the individual infectiousness parameters $\delta_i$ for cases were augmented variables. Therefore, we conducted our inference under a Bayesian framework using data augmentation Markov chain Monte Carlo (MCMC) algorithm to joint estimate the model parameters, the missing infection time, and augmented variables using metropolis-hasting algorithm. We separately fitted this model to identified studies. We assessed the model adequacy by comparing the observed and expected number of infections in households by household size. We compared the model with or without the random effect for variation of individual infectiousness by using deviance information criterion (DIC) (*Spiegelhalter et al., 2002*). Smaller DIC indicated a better model fit. DIC difference >5 was considered as substantial improvement (*Spiegelhalter et al., 2000*). Details of the model and inference could be found in Appendix 1. We conducted a sensitivity analysis that used a Gamma distribution, instead of exponential of normal distribution (Appendix 1).

## Meta-analysis and meta-regression

We conducted random effects meta-analyses on identified studies to obtain pooled estimates of individual infectiousness, using the inverse variance method and restricted maximum likelihood estimator for heterogeneity (*Hedges and Vevea, 1998*; *Langan et al., 2019*; *Thompson and Sharp, 1999*; *Veroniki et al., 2019*). Cochran Q test and the $I^2$ statistic were used to identify and quantify heterogeneity among included studies (*Cochran, 1954*; *Higgins et al., 2003*). An $I^2$ value more than 75% indicates high heterogeneity (*Higgins et al., 2003*). We conducted a meta-regression analysis to explore the association between infectiousness variation ($\sigma_{var}$, SD) of the distribution of number of secondary cases ($\sigma_{sec}$, and SAR), and further including the following factors: the mean and SD of number of household contacts, implementation of lockdown, ascertainment method of index and secondary cases, only ancestral strains are circulating in study period, and all household contacts were tested.

## Data availability

All data in this study are publicly available since they are extracted from published articles. Summarized data for analysis could be downloaded from https://github.com/timktsang/covid19_transmission_heterogeneity, (copy archived at swh:1:rev:634390fa66a4bfb998da691e7cd81cf45e38c6db; *Tsang, 2022*).

## Acknowledgements

The authors thank Maylis Layan for helpful discussion and Jiayi Zhou for technical assistance. This project was supported by the Health and Medical Research Fund, Food and Health Bureau, Government of the Hong Kong Special Administrative Region (grant no. COVID190118; BJC) and the Collaborative Research Fund (Project No. C7123-20G; BJC) of the Research Grants Council of the Hong Kong SAR Government. BJC is supported by the AIR@innoHK program of the Innovation and Technology Commission of the Hong Kong SAR Government. SC acknowledges financial support from the Investissement d'Avenir program, the Laboratoire d'Excellence Integrative Biology of Emerging Infectious Diseases program (grant ANR-10-LABX-62-IBEID), the EMERGEN project (ANRS0151), the INCEPTION project (PIA/ANR-16-CONV-0005), the European Union's Horizon 2020 research and innovation program under grant 101003589 (RECOVER) and 874735 (VEO), AXA and Groupama. TKT acknowledges the Seed Fund for Basic Research (202111159118) from the University of Hong Kong.

## Additional information

### Competing interests

Benjamin John Cowling: consults for AstraZeneca, Fosun Pharma, GlaxoSmithKline, Moderna, Pfizer, Roche and Sanofi Pasteur. The authors report no other potential conflicts of interest. The other authors declare that no competing interests exist.

### Funding

| Funder | Grant reference number | Author |
| --- | --- | --- |
| Food and Health Bureau | COVID190118 | Benjamin John Cowling |
| Research Grants Council, University Grants Committee | C7123-20G | Benjamin John Cowling |
| Innovation and Technology Commission of Hong Kong Special Administrative Government | AIR@innoHK | Benjamin John Cowling |
| Investissement d'Avenir | ANR-10-LABX-62-IBEID | Simon Cauchemez |
| EMERGEN | ANRS0151 | Simon Cauchemez |
| INCEPTION | PIA/ANR-16-CONV-0005 | Simon Cauchemez |

| Funder | Grant reference number | Author |
|---|---|---|
| AXA Research Fund | 101003589 (RECOVER) | Simon Cauchemez |
| Groupama | 874735 (VEO) | Simon Cauchemez |
| The University of Hong Kong | 202111159118 | Tim K Tsang |

The funders had no role in study design, data collection and interpretation, or the decision to submit the work for publication.

## Author contributions

Tim K Tsang, Conceptualization, Data curation, Software, Formal analysis, Funding acquisition, Validation, Investigation, Visualization, Methodology, Writing - original draft, Writing – review and editing; Xiaotong Huang, Data curation, Investigation, Visualization; Can Wang, Data curation, Investigation; Sijie Chen, Investigation, Visualization; Bingyi Yang, Writing – review and editing; Simon Cauchemez, Benjamin John Cowling, Conceptualization, Supervision, Funding acquisition, Methodology, Writing – review and editing

## Author ORCIDs

Tim K Tsang http://orcid.org/0000-0001-5037-6776
Simon Cauchemez http://orcid.org/0000-0001-9186-4549
Benjamin John Cowling http://orcid.org/0000-0002-6297-7154

## Decision letter and Author response

Decision letter https://doi.org/10.7554/eLife.82611.sa1
Author response https://doi.org/10.7554/eLife.82611.sa2

# Additional files

## Supplementary files

• MDAR checklist

## Data availability

The data and computer code (in R language) for conducting the data analysis can be downloaded from https://github.com/timktsang/covid19_transmission_heterogeneity (copy archived at swh:1:rev:634390fa66a4bfb998da691e7cd81cf45e38c6db).

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

# Appendix 1

## Systematic review

We conducted a systematic review following the Preferred Reporting Items for Systematic Review and Meta-analysis (PRISMA) statement (*Moher et al., 2009*). A standardized search was done in PubMed, Embase, and Web of Science, using the search term '("COVID-19" OR "SARS-CoV-2") AND ("household" OR "family") AND ("transmission")'. The search was done on 22 June, 2022, with no language restrictions. Additional relevant articles from the reference sections were also reviewed.

Two authors (XH and CW) independently screened the titles and extracted data from the included studies, with disagreement resolved by consensus together with a third author (TKT). Studies identified from different databases were de-duplicated.

## Household transmission model

Our aim was to explore the role of variation of individual infectiousness on transmission, with accounting for the variation and number of contacts, and the transmission probability. Therefore, we extended the household transmission model to account for variation of individual infectiousness of index cases (*Cauchemez et al., 2009*). We added a random effect on infectiousness of cases (*Tsang et al., 2014*). In the model, the hazard of infection of individual $j$ at time $t$ from an infected household member $i$, with infection time $t_i$ in household $k$, was

$$\lambda_{i \to j}(t) = \frac{\lambda_h}{X_k^{\beta}} * \exp(\delta_i) * f(t - t_i)$$

where $\lambda_h$ was the baseline hazard, $\delta_i$ was the relative individual infectiousness of cases, which followed a normal distribution with mean 0 and SD $\sigma_{var}$.

$X_k$ was the number of household contacts. $\beta$ was the parameter describing the relationship between number of household contacts and transmission rate. It ranged from 0 to 1, with 0 indicating that the transmission rate is independent of number of household contacts while 1 indicates that the transmission rate is inversely proportional to number of household contacts (i.e. dilution effect of the contact time per contact which is lower when the number of household contacts is higher). $f(\cdot)$ was the infectiousness profile since infection generated from the assumed incubation period (mean equal to 5 days) and infectious period (mean equal to 13) (*He et al., 2020*; *Jing et al., 2020*; *Li et al., 2020b*).

The daily hazard of infection from outside the household is assumed to be $\lambda_c(t) = \lambda_c$. Hence, the total hazard infection of individual $j$ on day $t$ was

$$\lambda_j(t) = \lambda_c + \sum_{i:\ infected} \lambda_{i \to j}(t)$$

Based on the transmission model, the probability that an individual $i$ was infected with infection time $t_{i0}$ was

$$P(y_i = 1, t_i = t_{i0}) = \left[1 - \exp(-\lambda_i(t_{i0}))\right] * \left[\exp\left(-\sum_{d=z_{i1}}^{t_{i0}-1} \lambda_i(d)\right)\right]$$

where $z_{i1}$ was the start of the follow-up day of individual $i$. Denote the total follow-up time for an individual $i$ as $z_{i2}$, then the probability that an individual $i$ did not get infected within the follow-up period was

$$P(y_i = 0, t_i = t_{i0} = z_{i2} + 1) = \exp\left(-\sum_{d=z_{i1}}^{t_{i0}-1} \lambda_i(d)\right)$$

Hence the log-likelihood function $L$ was

$$L = \sum_{i:\ y_i=1} \log\left(1 - \exp(-\lambda_i(t_{i0}))\right) - \sum_{i} \sum_{d=z_{i1}}^{t_{i0}-1} \lambda_i(d)$$

Index cases did not contribute to the likelihood and hence the summation was only on household contacts.

## Inference, model comparison, and validation

The infection time for cases were unavailable in the data extracted from publication. Also, the individual infectiousness parameter $\delta_i$ was an augmented variable. Therefore, we conducted our inference under a Bayesian framework with using data augmentation MCMC algorithm to joint estimate the model parameters, the missing infection time, and the relative individual infectiousness with using metropolis-hasting algorithm. For the parameter that only takes positive value, we used a vague Uniform(0,10) prior. For the parameter for relationship between number of household contacts and transmission, we used Uniform(0,1). The algorithm runs for 50,000 iterations after a burn-in of 50,000 iterations. Converge was visually assessed. One run takes about 3 hr on our typical desktop computer. We fitted the model to all studies, but only 14 of them could converge and provide robust estimates of individual infectiousness.

We assessed the model adequacy by comparing the observed and expected number of infections in households by household size. We simulated 10,000 datasets with a structure identical to that of the observed data (in terms of number of household contacts), with parameters randomly drawn from the posterior distribution of model parameter.

We compared the model with or without the random effect for variation of individual infectiousness by using DIC (*Spiegelhalter et al., 2002*). Smaller DIC indicated a better model fit. DIC difference >5 was considered as substantial improvement (*Spiegelhalter et al., 2000*). Since the likelihood of observed data could not be directly evaluated for a given model (*Celeux et al., 2006*), we used an importance sampling approach to estimate the likelihood for the observed data and evaluate the DIC (*Cauchemez et al., 2014*; *Liu, 2001*). For each household, we simulated 2000 datasets, with parameters randomly drawn from the posterior distribution. Then we compared the observed data and simulated data. The contribution to the likelihood of a household was equal to the proportion of simulated data with infection status that exactly matched the observed data, for all household members. To avoid the problem of 0-valued likelihood, we used the approach developed by *Cauchemez et al., 2014*, and assumed the sensitivity and specificity for diagnosing a case were both 99.99%.

## Comparison of modeling approach

Given that the infection time (or information that could inform infection time, such as symptom onset time) were not available for all studies, it may be possible to consider them as final size data, which may be analyzed by chain binomial model {Longini, 1988 #693;Longini, 1982 #335;Klick, 2011 #338;O'Neill, 2000 #368}. Including covariates affecting susceptibility is still possible, but showed to be difficult since the number of parameters increases exponentially by groups and type of infection {Klick, 2011 #338}. However, the model equation is about the escape probability of infection, which did not include the characteristic of infectee and therefore it is not possible to add covariates affecting infectiousness.

## Sensitivity analysis

Our model assumed that the individual infectiousness parameter $\delta_i$ followed a normal distribution, and it was multiplied to the hazard in the form of $\exp\left(\delta_i\right)$, therefore, the individual infectiousness essentially followed a lognormal distribution, which had a long tail. Therefore, we conducted a sensitivity analysis as follows:

$$\lambda_{i \to j}\left(t\right) = \frac{\lambda_h}{X_k^\beta} * \delta_i * f\left(t - t_i\right)$$

Here, $\delta_i$ followed a gamma distribution with mean 1, and SD $\sigma_{var}$, which had a shorter tail compared with lognormal distribution. Model comparison suggested that assuming gamma distribution for individual infectiousness parameter performed substantially worse, compared with the lognormal distribution in the main analysis (*Appendix 1—table 9*).

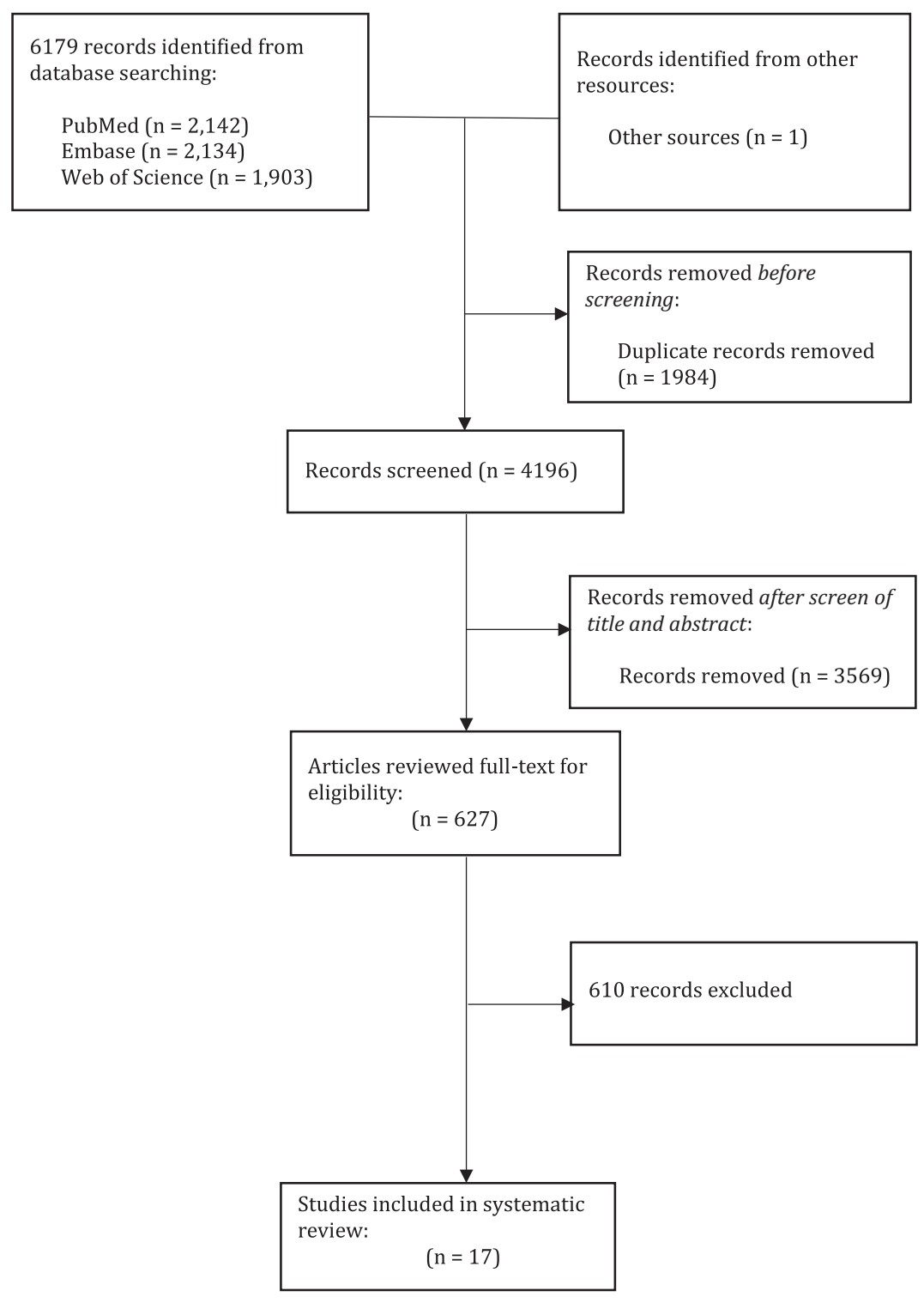

**Appendix 1—figure 1.** Process of systematic review.

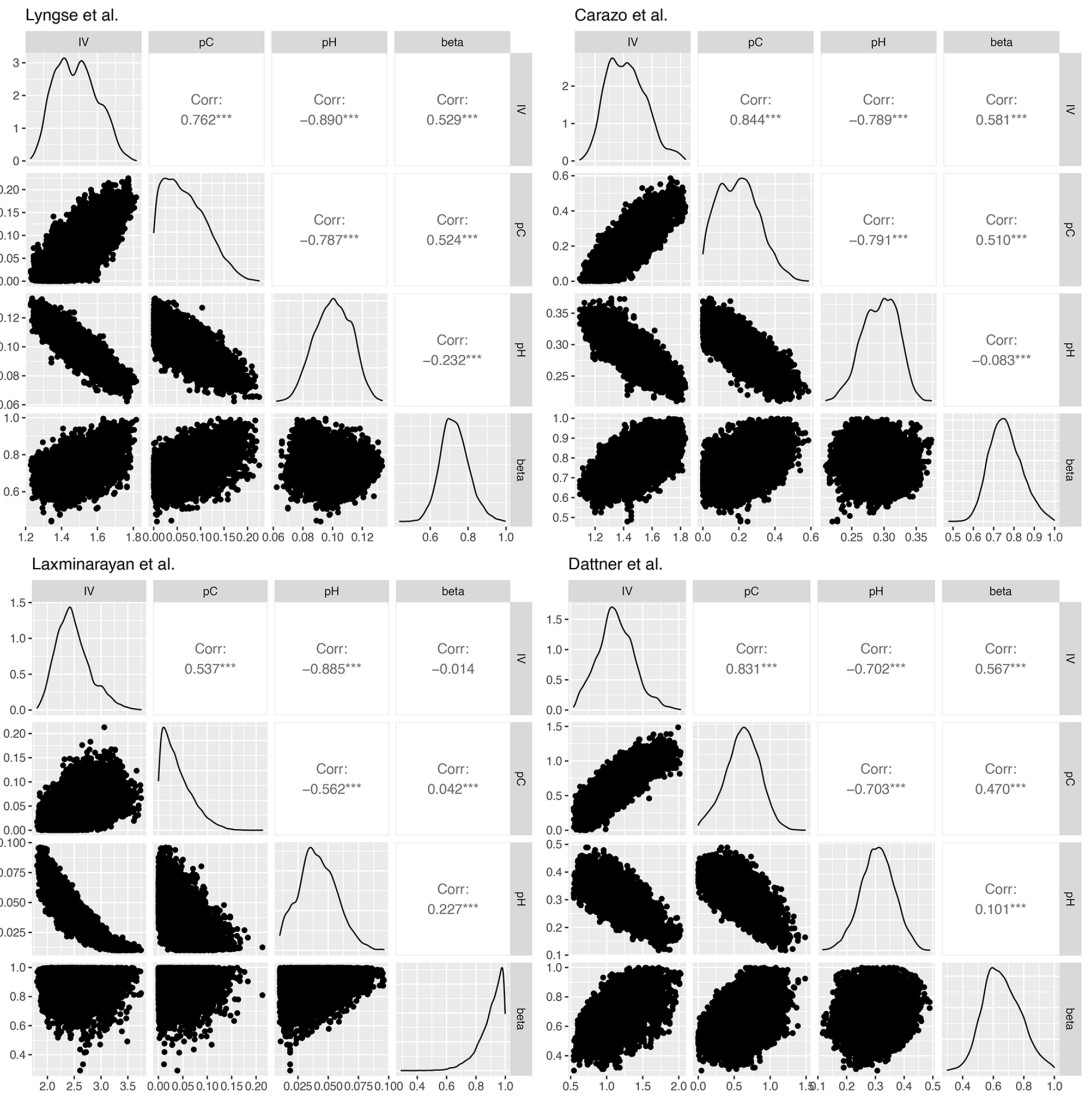

**Appendix 1—figure 2.** Correlation plots for the posterior distribution of model parameters in Lyngse et al., Carazo et al., Laxminarayan et al., and Dattner et al.

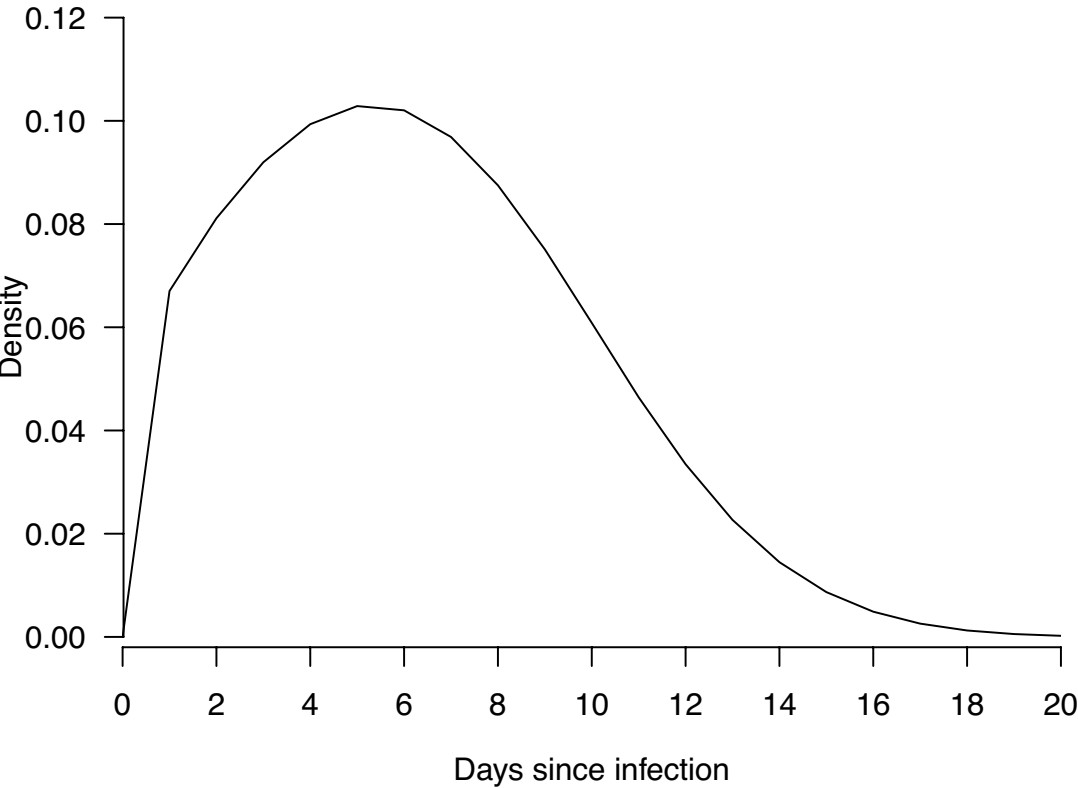

**Appendix 1—figure 3.** The density plot of the distribution of infectiousness profile since infections, used in the modeling analysis.

**Appendix 1—table 1.** Summary of characteristic of identified studies.

| Author (year) | Location | Study period | Case ascertainment method | Test coverage of identified contacts | SARS-COV-2 variant | Public health and social measures |
|---|---|---|---|---|---|---|
| *Lyngse et al., 2022* | Denmark | February 2020 to August 2020 | Index: RT-PCR Secondary: RT-PCR | All contacts were tested | Ancestral strains | Lockdown |
| *Carazo et al., 2022* | Canada | March 2020 to June 2020 | Index: RT-PCR Secondary: Symptom-based | NA | Ancestral strains | Handwashing, mask-wearing, and physical distancing |
| *Laxminarayan et al., 2020* | India | March 2020 to July 2020 | Index: RT-PCR Secondary: RT-PCR | All contacts were tested | Ancestral strains | Lockdown, social distancing, contact tracing |
| *Dattner et al., 2021* | Israel | March 2020 to June 2020 | Index: RT-PCR and Serology Secondary: RT-PCR and Serology | All contacts were tested | Ancestral strains | Lockdown |
| *Layan et al., 2022* | Israel | December 2020 to April 2021 | Index: RT-PCR Secondary: RT-PCR | All contacts were tested | Ancestral strains, alpha | Vaccination |
| *Hart et al., 2022* | UK | March 2020 to November 2020 | Index: RT-PCR Secondary: RT-PCR and antibody test | All contacts were tested | Ancestral strains | Isolation |
| *Hubiche et al., 2021* | Canada | April 2020 to June 2020 | Index: Symptom-based and RT-PCR and serology Secondary: Symptom-based and RT-PCR and serology | NA | Ancestral strains | Closure of school |
| *Wilkinson et al., 2021* | Manitoba, Canada | Mid-January 2020 to late March 2020 | Index: NAAT assay Secondary: NAAT assay | Only symptomatic contacts were tested | Ancestral strains | Isolation and contact tracing |

*Appendix 1—table 1 Continued on next page*

*Appendix 1—table 1 Continued*

| Author (year) | Location | Study period | Case ascertainment method | Test coverage of identified contacts | SARS-COV-2 variant | Public health and social measures |
|---|---|---|---|---|---|---|
| *Tsang et al., 2022* | Shandong Province, China | January 2020 to May 2020 | Index: RT-PCR Secondary: RT-PCR | All contacts were tested | Ancestral strains | Isolation, mask-waring, social distancing |
| *Reukers et al., 2021* | Netherlands | March 2020 to May 2020 | Index: RT-PCR Secondary: RT-PCR | NA | Ancestral strains | Social distancing |
| *Han et al., 2022* | Netherlands | March 2020 to April 2020 | Index: RT-PCR and serology Secondary: RT-PCR and serology | All contacts were tested | Ancestral strains | Social distancing, self-quarantine and self-isolation orders, closure of schools, bars and restaurants, and urging people to work from home |
| *Méndez-Echevarría et al., 2021* | Madrid, Spain | March 2020 to May 2020 | Index: RT-PCR and serology Secondary: RT-PCR and serology | All contacts were tested | Ancestral strains | Lockdown |
| *Dutta et al., 2020* | Rajasthan, India | May 2020 to July 2020 | Index: RT-PCR Secondary: RT-PCR | All contacts were tested | Ancestral strains | Lockdown and stay-at-home orders, physical distancing |
| *Koureas et al., 2021* | Greece | April 2020 to June 2020 | Index: RT-PCR Secondary: RT-PCR | All contacts were tested | Ancestral strains | Quarantine, screening, movement restrictions and gathering prohibition, isolation |
| *Bernardes-Souza et al., 2021; Méndez-Echevarría et al., 2021* | Brazil | May 2020 to June 2020 | Index: Serology and RT-PCR Secondary: Serology | All contacts were tested | Ancestral strains | Lockdown, gathering restrictions and face mask mandates |
| *Posfay-Barbe et al., 2020* | Switzerland | March 2020 to April 2020 | Index: RT-PCR Secondary: RT-PCR | NA | Ancestral strains | Closure of schools, daycares, restaurants, bars, and shops, social distancing |
| *Hsu et al., 2021* | Taiwan, China | January 2020 to February 2021 | Index: RT-PCR Secondary: RT-PCR | Only symptomatic contacts were tested | Ancestral strains, alpha | Mask-wearing, hand washing, and social distancing |

**Appendix 1—table 2.** Summary of model estimates.

| Article | Estimates of infectiousness variation | Estimates of probability of infection from community ($10^{-2}$) | Estimates of probability of infection from household | Relationship between transmission and number of contacts ($\beta$ |
|---|---|---|---|---|
| Lyngse et al. | 1.48 (1.29, 1.7) | 0.06 (0, 0.16) | 0.1 (0.08, 0.12) | 0.72 (0.59, 0.89) |
| Carazo et al. | 1.41 (1.19, 1.72) | 0.2 (0.01, 0.43) | 0.3 (0.24, 0.34) | 0.75 (0.62, 0.93) |
| Laxminarayan et al. | 2.44 (1.98, 3.23) | 0.03 (0, 0.11) | 0.04 (0.01, 0.07) | 0.92 (0.69, 1) |
| Dattner et al. | 1.12 (0.65, 1.69) | 0.63 (0.12, 1.06) | 0.31 (0.19, 0.42) | 0.65 (0.45, 0.91) |
| Layan et al. | 1.12 (0.74, 1.76) | 0.3 (0.02, 0.96) | 0.26 (0.15, 0.39) | 0.2 (0.01, 0.6) |
| Hart et al. | 0.35 (0.11, 0.96) | 0.74 (0.05, 2.12) | 0.51 (0.32, 0.65) | 0.72 (0.32, 0.98) |
| Hubiche et al. | 1.03 (0.45, 2.68) | 1.24 (0.14, 2.48) | 0.32 (0.06, 0.57) | 0.67 (0.13, 0.98) |
| Wilkinson et al. | 1.05 (0.12, 3.51) | 0.32 (0.02, 0.91) | 0.07 (0, 0.18) | 0.43 (0.02, 0.97) |
| Tsang et al. | 2.83 (1.48, 4.73) | 0.45 (0.05, 1.03) | 0.06 (0, 0.26) | 0.66 (0.06, 0.99) |
| Reukers et al. | 1.79 (0.72, 4.5) | 1.53 (0.18, 2.93) | 0.22 (0.01, 0.6) | 0.67 (0.07, 0.98) |
| Han et al. | 1.71 (0.51, 4.13) | 1.67 (0.19, 2.99) | 0.21 (0.02, 0.61) | 0.69 (0.07, 0.99) |
| Méndez-Echevarría et al. | 0.51 (0.13, 1.7) | 0.91 (0.05, 2.53) | 0.28 (0.06, 0.52) | 0.57 (0.04, 0.98) |
| Dutta et al. | 1.53 (0.5, 4.2) | 1.1 (0.12, 2.26) | 0.24 (0.01, 0.63) | 0.78 (0.15, 0.99) |

*Appendix 1—table 2 Continued on next page*

*Appendix 1—table 2 Continued*

| Article | Estimates of infectiousness variation | Estimates of probability of infection from community ($10^{-2}$) | Estimates of probability of infection from household | Relationship between transmission and number of contacts ($\beta$ |
|---|---|---|---|---|
| Koureas et al. | 1.88 (0.6, 3.66) | 0.84 (0.1, 1.76) | 0.13 (0.01, 0.42) | 0.44 (0.02, 0.96) |

**Appendix 1—table 3.** Model adequacy check for Lyngse et al.
Each element of the table has the format observed frequency – expected (posterior mean) frequency (95% credible interval).

| Number of household contacts | Number of infected household contacts | | | | | |
|---|---|---|---|---|---|---|
| | 0 | 1 | 2 | 3 | 4 | 5 |
| 1 | 2366–2377 (2319, 2433) | 569–558 (502, 616) | NA | NA | NA | NA |
| 2 | 1117–1105 (1070, 1138) | 227–227 (198, 259) | 77–88 (69, 109) | NA | NA | NA |
| 3 | 1135–1118 (1077, 1156) | 214–235 (203, 267) | 89–86 (67, 106) | 41–41 (27, 56) | NA | NA |
| 4 | 521–528 (501, 555) | 119–115 (94, 137) | 40–42 (30, 56) | 25–19 (11, 30) | 11–10 (4, 18) | NA |
| 5 | 161–167 (152, 181) | 42–38 (27, 50) | 14–14 (7, 22) | 7–7 (2, 12) | 7–3 (0, 8) | 0–2 (0, 5) |

**Appendix 1—table 4.** Model adequacy check for Carazo et al.
Each element of the table has the format observed frequency – expected (posterior mean) frequency (95% credible interval).

| Number of household contacts | Number of infected household contacts | | | | | |
|---|---|---|---|---|---|---|
| | 0 | 1 | 2 | 3 | 4 | 5 |
| 1 | 803–814 (765, 862) | 532–521 (473, 570) | NA | NA | NA | NA |
| 2 | 476–454 (423, 485) | 179–186 (160, 212) | 154–169 (144, 195) | NA | NA | NA |
| 3 | 518–520 (483, 557) | 201–202 (175, 232) | 132–130 (107, 155) | 133–131 (106, 158) | NA | NA |
| 4 | 217–221 (197, 245) | 85–87 (70, 106) | 52–52 (39, 67) | 43–39 (27, 52) | 45–41 (28, 56) | NA |
| 5 | 76–74 (61, 88) | 30–30 (20, 41) | 20–18 (10, 26) | 13–12 (6, 20) | 7–10 (5, 17) | 10–11 (5, 18) |

**Appendix 1—table 5.** Model adequacy check for Laxminarayan et al.
Each element of the table has the format observed frequency – expected (posterior mean) frequency (95% credible interval).

| Number of household contacts | Number of infected household contacts | | | | | | | | | |
|---|---|---|---|---|---|---|---|---|---|---|
| | 0 | 1 | 2 | 3 | 4 | 5 | 6 | 7 | 8 | 9 |
| 1 | 124–134 (122, 144) | 37–27 (17, 39) | NA | NA | NA | NA | NA | NA | NA | NA |
| 2 | 135–137 (125, 147) | 12–18 (10, 27) | 20–12 (6, 20) | NA | NA | NA | NA | NA | NA | NA |
| 3 | 188–176 (163, 188) | 13–22 (13, 33) | 11–9 (4, 16) | 5–9 (4, 17) | NA | NA | NA | NA | NA | NA |
| 4 | 127–118 (108, 128) | 11–15 (8, 23) | 5–5 (1, 11) | 3–3 (0, 7) | 1–4 (1, 10) | NA | NA | NA | NA | NA |
| 5 | 75–76 (67, 84) | 12–10 (4, 16) | 3–3 (0, 8) | 1–2 (0, 5) | 1–1 (0, 4) | 3–2 (0, 6) | NA | NA | NA | NA |
| 6 | 41–40 (34, 46) | 7–5 (1, 10) | 2–2 (0, 5) | 0–1 (0, 3) | 0–0 (0, 3) | 0–0 (0, 2) | 1–1 (0, 3) | NA | NA | NA |
| 7 | 32–31 (25, 36) | 3–4 (1, 9) | 2–1 (0, 4) | 1–1 (0, 3) | 1–0 (0, 2) | 0–0 (0, 2) | 0–0 (0, 2) | 0–0 (0, 3) | NA | NA |
| 8 | 10–13 (10, 16) | 2–2 (0, 5) | 3–0 (0, 2) | 1–0 (0, 2) | 0–0 (0, 1) | 1–0 (0, 1) | 0–0 (0, 1) | 0–0 (0, 1) | 0–0 (0, 1) | NA |
| 9 | 14–16 (12, 20) | 2–2 (0, 6) | 1–1 (0, 3) | 2–0 (0, 2) | 1–0 (0, 2) | 0–0 (0, 1) | 0–0 (0, 1) | 0–0 (0, 1) | 0–0 (0, 1) | 1–0 (0, 2) |

**Appendix 1—table 6.** Model adequacy check for Dattner et al.
Each element of the table has the format observed frequency – expected (posterior mean) frequency (95% credible interval).

| Number of household contacts | Number of infected household contacts | | | | | | | | | |
|---|---|---|---|---|---|---|---|---|---|---|
| | 0 | 1 | 2 | 3 | 4 | 5 | 6 | 7 | 8 | 9 |
| 1 | 85–93 (77, 108) | 73–65 (50, 81) | NA | NA | NA | NA | NA | NA | NA | NA |
| 2 | 46–42 (32, 52) | 21–25 (17, 34) | 19–19 (11, 28) | NA | NA | NA | NA | NA | NA | NA |
| 3 | 38–29 (21, 38) | 13–18 (11, 25) | 9–12 (6, 19) | 8–9 (4, 15) | NA | NA | NA | NA | NA | NA |
| 4 | 24–24 (16, 32) | 19–15 (9, 23) | 11–10 (5, 17) | 8–7 (3, 13) | 1–5 (1, 11) | NA | NA | NA | NA | NA |
| 5 | 25–21 (14, 30) | 15–15 (9, 22) | 12–10 (5, 17) | 7–7 (3, 13) | 3–5 (2, 10) | 2–4 (1, 9) | NA | NA | NA | NA |
| 6 | 13–16 (9, 23) | 19–12 (6, 18) | 3–8 (3, 14) | 5–6 (2, 11) | 3–5 (1, 9) | 5–3 (0, 7) | 5–2 (0, 6) | NA | NA | NA |
| 7 | 5–9 (4, 15) | 6–7 (3, 13) | 5–5 (2, 10) | 7–4 (1, 8) | 4–3 (0, 7) | 4–2 (0, 6) | 4–2 (0, 5) | 0–1 (0, 4) | NA | NA |
| 8 | 10–8 (3, 14) | 7–7 (3, 12) | 3–5 (1, 10) | 4–4 (1, 8) | 0–3 (0, 7) | 3–2 (0, 6) | 3–2 (0, 5) | 4–1 (0, 4) | 0–1 (0, 4) | NA |
| 9 | 6–6 (2, 12) | 3–6 (2, 10) | 4–4 (1, 9) | 6–3 (0, 7) | 4–3 (0, 6) | 1–2 (0, 5) | 2–2 (0, 5) | 2–1 (0, 4) | 2–1 (0, 3) | 0–0 (0, 3) |

**Appendix 1—table 7.** Comparison of model estimates from 100k Markov chain Monte Carlo (MCMC) iterations and 500k MCMC iterations.

| Article | Number of MCMC iterations | Estimates of infectiousness variation | Estimates of probability of infection from community (10⁻²) | Estimates of probability of infection from household | Relationship between transmission and number of contacts ($\beta$ |
|---|---|---|---|---|---|
| Lyngse et al. | 100,000 | 1.48 (1.29, 1.7) | 0.06 (0, 0.16) | 0.11 (0.08, 0.13) | 0.72 (0.59, 0.89) |
| | 500,000 | 1.49 (1.29, 1.84) | 0.07 (0, 0.2) | 0.1 (0.07, 0.13) | 0.74 (0.6, 0.92) |
| Carazo et al. | 100,000 | 1.41 (1.19, 1.72) | 0.2 (0.01, 0.43) | 0.35 (0.27, 0.42) | 0.75 (0.62, 0.93) |
| | 500,000 | 1.44 (1.19, 1.78) | 0.21 (0.02, 0.46) | 0.34 (0.26, 0.41) | 0.77 (0.62, 0.95) |
| Laxminarayan et al. | 100,000 | 2.44 (1.98, 3.23) | 0.03 (0, 0.11) | 0.04 (0.01, 0.08) | 0.92 (0.69, 1) |
| | 500,000 | 2.47 (1.99, 3.23) | 0.03 (0, 0.11) | 0.04 (0.01, 0.07) | 0.92 (0.67, 1) |
| Dattner et al. | 100,000 | 1.12 (0.65, 1.69) | 0.63 (0.12, 1.06) | 0.37 (0.21, 0.54) | 0.65 (0.45, 0.91) |
| | 500,000 | 1.06 (0.65, 1.55) | 0.57 (0.1, 1) | 0.38 (0.24, 0.54) | 0.63 (0.43, 0.88) |

**Appendix 1—table 8.** Simulation study for validating the estimates and the corresponding power.

| Parameter | Simulation value | Mean estimate | Proportion covered (over 50) |
|---|---|---|---|
| Lyngse et al. | | | |
| $\sigma_{var}$: Infectiousness variation | 1.48 | 1.57 | 0.94 |
| $\lambda_c$: hazard of infection from outside the household (10⁻²) | 0.06 | 0.12 | 0.88 |
| $\lambda_h$: hazard of infection from an infected household member | 0.11 | 0.09 | 0.92 |
| $\beta$: relationship between number of household contacts and transmission rate | 0.72 | 0.75 | 0.96 |
| Carazo et al. | | | |
| $\sigma_{var}$: Infectiousness variation | 1.41 | 1.49 | 0.94 |
| $\lambda_c$: hazard of infection from outside the household (10⁻²) | 0.2 | 0.24 | 0.96 |
| $\lambda_h$: hazard of infection from an infected household member | 0.35 | 0.33 | 0.92 |
| $\beta$: relationship between number of household contacts and transmission rate | 0.75 | 0.77 | 0.96 |

*Appendix 1—table 8 Continued on next page*

*Appendix 1—table 8 Continued*

| Parameter | Simulation value | Mean estimate | Proportion covered (over 50) |
|---|---|---|---|
| Laxminarayan et al. | | | |
| $\sigma_{var}$: Infectiousness variation | 2.44 | 2.6 | 0.94 |
| $\lambda_c$: hazard of infection from outside the household ($10^{-2}$) | 0.03 | 0.05 | 0.96 |
| $\lambda_h$: hazard of infection from an infected household member | 0.04 | 0.03 | 0.9 |
| $\beta$: relationship between number of household contacts and transmission rate | 0.92 | 0.82 | 1 |
| Dattner et al. | | | |
| $\sigma_{var}$: Infectiousness variation | 1.12 | 1.18 | 0.92 |
| $\lambda_c$: hazard of infection from outside the household ($10^{-2}$) | 0.63 | 0.69 | 0.92 |
| $\lambda_h$: hazard of infection from an infected household member | 0.37 | 0.36 | 0.96 |
| $\beta$: relationship between number of household contacts and transmission rate | 0.65 | 0.68 | 0.98 |

**Appendix 1—table 9.** A sensitivity analysis of using normal distribution (main analysis) instead of lognormal distribution (sensitivity analysis) for individual infectiousness.

| Article | Model | Estimates of infectiousness variation | Estimates of probability of infection from community ($10^{-2}$) | Estimates of probability of infection from household | Relationship between transmission and number of contacts ($\beta$ | Difference in DIC |
|---|---|---|---|---|---|---|
| Lyngse et al. | Main analysis | 1.48 (1.29, 1.7) | 0.06 (0, 0.16) | 0.11 (0.08, 0.13) | 0.72 (0.59, 0.89) | |
| | Sensitivity analysis | 0.93 (0.84, 1.03) | 0.03 (0, 0.11) | 0.06 (0.05, 0.07) | 0.7 (0.58, 0.83) | 2821 |
| Carazo et al. | Main analysis | 1.41 (1.19, 1.72) | 0.2 (0.01, 0.43) | 0.35 (0.27, 0.42) | 0.75 (0.62, 0.93) | |
| | Sensitivity analysis | 1.05 (0.92, 1.21) | 0.06 (0, 0.24) | 0.18 (0.16, 0.2) | 0.71 (0.59, 0.84) | 2569 |
| Laxminarayan et al. | Main analysis | 2.44 (1.98, 3.23) | 0.03 (0, 0.11) | 0.04 (0.01, 0.08) | 0.92 (0.69, 1) | |
| | Sensitivity analysis | 1.31 (1.14, 1.48) | 0.02 (0, 0.08) | 0.05 (0.03, 0.06) | 0.91 (0.66, 1) | 818 |
| Dattner et al. | Main analysis | 1.12 (0.65, 1.69) | 0.63 (0.12, 1.06) | 0.37 (0.21, 0.54) | 0.65 (0.45, 0.91) | |
| | Sensitivity analysis | 0.66 (0.47, 0.88) | 0.3 (0.02, 0.71) | 0.18 (0.13, 0.22) | 0.56 (0.4, 0.75) | 908 |

**Appendix 1—table 10.** Summary of characteristic of identified studies.
SD: standard deviation.

| Article | Number of households | Number of contacts | Number of secondary cases | Mean number of contacts | SD of number of contact | Secondary attack rate (SAR) | SD of number of secondary cases ($\sigma_{sec}$) |
|---|---|---|---|---|---|---|---|
| Lyngse et al. | 6782 | 14,233 | 1902 | 2.1 (2.07, 2.13) | 1.17 (1.15, 1.19) | 0.13 (0.13, 0.14) | 0.6 (0.59, 0.61) |
| Carazo et al. | 3727 | 8460 | 2574 | 2.27 (2.23, 2.31) | 1.19 (1.16, 1.21) | 0.3 (0.29, 0.31) | 0.97 (0.95, 0.99) |
| Laxminarayan et al. | 915 | 3113 | 283 | 3.4 (3.28, 3.53) | 1.94 (1.86, 2.04) | 0.09 (0.08, 0.1) | 0.81 (0.77, 0.85) |
| Dattner et al. | 591 | 2211 | 720 | 3.74 (3.54, 3.94) | 2.5 (2.36, 2.65) | 0.33 (0.31, 0.35) | 1.6 (1.51, 1.7) |
| Layan et al. | 208 | 670 | 264 | 3.22 (3.01, 3.44) | 1.58 (1.45, 1.75) | 0.39 (0.36, 0.43) | 1.59 (1.45, 1.76) |
| Hart et al. | 172 | 433 | 194 | 2.52 (2.34, 2.69) | 1.16 (1.05, 1.29) | 0.45 (0.4, 0.5) | 1.11 (1, 1.24) |
| Hubiche et al. | 103 | 291 | 119 | 2.83 (2.6, 3.05) | 1.18 (1.04, 1.37) | 0.41 (0.35, 0.47) | 1.14 (1, 1.32) |
| Wilkinson et al. | 95 | 220 | 26 | 2.32 (2.02, 2.61) | 1.47 (1.28, 1.71) | 0.12 (0.08, 0.17) | 0.66 (0.58, 0.77) |

*Appendix 1—table 10 Continued on next page*

*Appendix 1—table 10 Continued*

| Article | Number of households | Number of contacts | Number of secondary cases | Mean number of contacts | SD of number of contact | Secondary attack rate (SAR) | SD of number of secondary cases ($\sigma_{sec}$) |
|---|---|---|---|---|---|---|---|
| Tsang et al. | 47 | 189 | 38 | 4.02 (3.39, 4.66) | 2.22 (1.85, 2.79) | 0.2 (0.15, 0.27) | 1.31 (1.09, 1.65) |
| Reukers et al. | 55 | 187 | 78 | 3.4 (3.11, 3.69) | 1.1 (0.93, 1.35) | 0.42 (0.35, 0.49) | 1.33 (1.12, 1.64) |
| Han et al. | 55 | 185 | 78 | 3.36 (3.08, 3.65) | 1.08 (0.91, 1.33) | 0.42 (0.35, 0.5) | 1.27 (1.07, 1.57) |
| Méndez-Echevarría et al. | 63 | 174 | 57 | 2.76 (2.57, 2.95) | 0.78 (0.66, 0.94) | 0.33 (0.26, 0.4) | 0.96 (0.82, 1.17) |
| Dutta et al. | 32 | 170 | 55 | 5.31 (4.52, 6.1) | 2.28 (1.83, 3.03) | 0.32 (0.25, 0.4) | 1.59 (1.28, 2.12) |
| Koureas et al. | 32 | 153 | 50 | 4.78 (3.94, 5.62) | 2.43 (1.95, 3.23) | 0.33 (0.25, 0.41) | 2.14 (1.72, 2.84) |
| Bernardes-Souza et al. | 40 | 112 | 55 | 2.8 (2.33, 3.27) | 1.51 (1.23, 1.93) | 0.49 (0.4, 0.59) | 1.48 (1.21, 1.9) |
| Posfay-Barbe et al. | 39 | 111 | 46 | 2.85 (2.54, 3.16) | 0.99 (0.81, 1.27) | 0.41 (0.32, 0.51) | 0.94 (0.77, 1.21) |
| Hsu et al. | 38 | 96 | 49 | 2.53 (2.12, 2.93) | 1.27 (1.03, 1.64) | 0.51 (0.41, 0.61) | 0.87 (0.71, 1.12) |

**Appendix 1—table 11.** Association between infectiousness variation estimated from household transmission models, and other statistics from 17 household studies, based on meta-regression.

| Statistic | Infectiousness variation ($\sigma_{var}$*) | Standard deviation (SD) of the distribution of number of secondary cases ($\sigma_{sec}$) | Secondary attack rate (SAR) |
|---|---|---|---|
| Pooled estimates | 1.33 (0.95, 1.70) | 1.19 (1.03, 1.35) | 0.35 (0.28, 0.44) |
| $I^2$ | 78.4 | 99.2 | 99.1 |
| Factors | | | |
| Estimate | $\beta$ | $\beta$ | exp ($\beta$) |
| Mean number of contacts | 0.31 (−0.16, 0.78) | **0.35 (0.22, 0.48)** | 1.09 (0.83, 1.43) |
| SD number of contacts | 0.55 (−0.12, 1.21) | **0.42 (0.15, 0.69)** | 0.80 (0.51, 1.25) |
| Implementation of lockdown | 0.16 (−0.61, 0.93) | −0.06 (−0.40, 0.28) | 0.69 (0.43, 1.10) |
| Index cases are confirmed by PCR only | 0.48 (−0.32, 1.29) | 0.02 (−0.33, 0.38) | 1.09 (0.65, 1.85) |
| Secondary cases are confirmed by PCR only | **0.78 (0.13, 1.43)** | 0.03 (−0.30, 0.35) | 0.88 (0.55, 1.41) |
| Only ancestral strains are circulating in study period | 0.76 (−0.03, 1.55) | 0.07 (−0.31, 0.45) | 0.75 (0.43, 1.30) |
| All household contacts were tested | 0.03 (−0.87, 0.93) | 0.28 (−0.04, 0.59) | 0.96 (0.58, 1.58) |

*Estimates based on results from 14 studies

**Appendix 1—table 12.** Pooled estimates for duration of viral shedding for COVID-19 from 11 systematic reviews.

| Study | Sampling site | Laboratory method | Pooled estimates (mean/median) | 95% CI | Range | $I^2$ |
|---|---|---|---|---|---|---|
| Cevik | Upper respiratory tract | PCR | 17.0 | 15.5–18.6 | 6.0–53·9 | |
| Cevik | Lower respiratory tract | PCR | 14.6 | 9.3–20.0 | 6.2–22.7 | 97% |
| Cevik | Stool | Viral culture | 17.2 | 14.4–20.1 | 9.8–27.9 | 96.6% |
| Cevik | Serum/blood | Viral culture | 16.6 | 3.6–29.7 | 10.0–23.3 | 99% |
| Fontana | Respiratory sources | PCR | 18.4* | 15.5–21.3 | 5.5–53.5 | 98.87% |
| Fontana | Rectal/stool | PCR | 22.1* | 14.4–29.8 | 11–33 | 95.86% |

*Appendix 1—table 12 Continued on next page*

*Appendix 1—table 12 Continued*

| Study | Sampling site | Laboratory method | Pooled estimates (mean/median) | 95% CI | Range | I² |
|---|---|---|---|---|---|---|
| Okita | Upper respiratory tract (nasal swab+throat swab) | PCR | 18.29 | 17.00–19.89 | 8.33–39.97 | 99% |
| Okita | Sputum | PCR | 23.79 | 20.43–27.16 | 15.50–32.00 | 93% |
| Okita | Blood | PCR | 14.60 | 12.16–17.05 | 11.00–17.58 | 88% |
| Okita | Stool | PCR | 22.38 | 18.40–26.35 | 10.67–51.40 | 97% |
| Qutub | Respiratory tract | Viral culture | 28.75/11** | 8.5–14.5 | | |
| Rahmani | Respiratory tract | PCR | 27.90 | 23.27–32.52 | 7.40–132.00 | 99.1% |
| Xu | Respiratory tract (symptomatic cases) | PCR | 11.1±5.8*** | | 0–24 | |
| Xu | Gastrointestinal tract (symptomatic cases) | PCR | 23.6±8.8*** | | 10–33 | |
| Yan | Unrestricted | PCR | 16.8 | 14.8–19.4 | | 99.56% |
| Yan | Stool | PCR | 30.3 | 23.1–39.2 | | 92.09% |
| Yan | Respiratory tract | PCR | 17.5 | 14.9–20.6 | | 99.67% |
| Yan | Upper respiratory tract | PCR | 17.5 | 14.6–21.0 | | 99.53% |
| Diaz | Stool | PCR | 22* | 19–25 | | |
| Chen | (Asymptomatic infections) | | 14.14* | 11.25–17.04 | 11.00–17.25 | |
| Li | Upper respiratory tract (nasopharyngeal/throat swabs) | PCR | 11.43 | 10.1–12.77 | | 75.3% |
| Zhang | Stool | PCR | 21.8 | 16.4–27.1 | | |
| Zhang | Respiratory tract | PCR | 14.7 | 9.9–19.5 | | |

*Median estimate, **median/grouped median, ***this study analyzed by cases, and reported mean ± SD.

**Appendix 1—table 13.** Pooled estimates for duration of viral shedding for COVID-19 in subgroups from seven systematic reviews.

| Study | Sampling site (subgroups) | Laboratory method | Pooled estimates (mean/median) | 95% CI | Range | I² |
|---|---|---|---|---|---|---|
| Fontana | Respiratory sources (among severely ill patients) | PCR | 19.8* | 16.2–23.5 | 11–31 | 96.42% |
| Fontana | Respiratory sources (in mild-to-moderate illness) | PCR | 17.2* | 14.0–20.5 | 8–25 | 95.64% |
| Okita | Upper respiratory tract (nasal swab) | PCR | 19.34 | 16.60–22.07 | | 99% |
| Okita | Upper respiratory tract (throat swab) | PCR | 17.85 | 16.43–19.26 | | 98% |
| Okita | Upper respiratory tract (age < 60) | PCR | 16.95 | 13.56–20.35 | 8.62–35.67 | 98% |
| Okita | Upper respiratory tract (age ≥ 60) | PCR | 21.24 | 14.06–28.41 | 8.25–40.33 | 99% |
| Okita | Upper respiratory tract (with comorbidities) | PCR | 20.26 | 17.60–22.92 | 9.67–34.00 | 93% |
| Okita | Upper respiratory tract (without comorbidities) | PCR | 14.66 | 12.63–16.69 | 10.82–27.25 | 85% |
| Okita | Upper respiratory tract (severe patients) | PCR | 20.79 | 18.03–23.55 | 14.00–38.33 | 98% |
| Okita | Upper respiratory tract (nonsevere patients) | PCR | 16.36 | 14.07–18.66 | 8.00–37.33 | 99% |

*Appendix 1—table 13 Continued on next page*

*Appendix 1—table 13 Continued*

| Study | Sampling site (subgroups) | Laboratory method | Pooled estimates (mean/ median) | 95% CI | Range | $I^2$ |
|---|---|---|---|---|---|---|
| Okita | Upper respiratory tract (severe patients) (for studies with mean age ≥ 40 + comorbidity > 30%) | PCR | 21.53 | 17.57–25.50 | 14.00–29.50 | 91% |
| Okita | Upper respiratory tract (severe patients) (for studies with mean age ≥ 40 + comorbidity >30%) | PCR | 20.08 | 15.87–24.29 | 13.12–33.67 | 100% |
| Okita | Upper respiratory tract (treated with glucocorticoid) | PCR | 19.72 | 17.92–21.52 | 13.87–33.67 | 92% |
| Okita | Upper respiratory tract (treated without glucocorticoid) | PCR | 15.64 | 14.18–17.10 | 8.33–31.60 | 96% |
| Okita | Upper respiratory tract (treated with glucocorticoid) (for studies with mean age: 30–60+comorbidity > 50%) | PCR | 21.98 | 16.48–27.48 | 14.25–33.67 | 94% |
| Okita | Upper respiratory tract (treated without glucocorticoid) (for studies with mean age: 30–60+comorbidity > 50%) | PCR | 16.14 | 12.60–19.68 | 13.22–24.44 | 92% |
| Okita | Upper respiratory tract (Asian) | PCR | 18.10 | 16.95–19.25 | 8.33–34.75 | 98% |
| Okita | Upper respiratory tract (European) | PCR | 19.27 | 11.59–26.95 | 8.50–39.97 | 100% |
| Okita | Upper respiratory tract (Asian) (for studies with mean age ≥ 40 + comorbidity >40%) | PCR | 20.66 | 18.18–23.14 | 12.00–32.00 | 96% |
| Okita | Upper respiratory tract (European) (for studies with mean age ≥ 40 + comorbidity > 40%) | PCR | 23.68 | 10.85–36.51 | 13.00–39.97 | 100% |
| Qutub | Respiratory tract (severe patients) | Viral culture | 47.5/20** | 9.0–53.0 | | |
| Qutub | Respiratory tract (severe patients were not specified of excluded) | Viral culture | 10/9** | 8.0–13.0 | | |
| Qutub | Respiratory tract (immunocompromised patients) | Viral culture | 54.36/20** | 9.0–85.98 | | |
| Qutub | Respiratory tract (immunocompromised patients were not specified of excluded) | Viral culture | 11.67/9** | 8.2–13.3 | | |
| Rahmani | Not specific (immunocompetent individuals) | PCR | 26.54 | 21.44–31.64 | 7.40–91.20 | 99.3% |
| Rahmani | Not specific (immunocompromised individuals) | PCR | 36.28 | 21.93–50.63 | 15.90–132.00 | 94.2% |
| Xu | Respiratory tract (asymptomatic cases) | PCR | 9.4±5.1*** | | | |
| Xu | Gastrointestinal tract (asymptomatic cases) | PCR | 16.8±9.8*** | | | |
| Yan | Unrestricted (symptomatic cases) | PCR | 19.7 | 17.2–22.7 | | 99.34% |
| Yan | Unrestricted (asymptomatic cases) | PCR | 10.9 | 8.3–14.3 | | 98.89% |
| Yan | Unrestricted (severe patients) | PCR | 24.3 | 18.9–31.1 | | 91.88% |
| Yan | Unrestricted (nonsevere patients) | PCR | 22.8 | 16.4–32.0 | | 99.81% |
| Yan | Unrestricted (females) | PCR | 19.4 | 9.5–39.4 | | 93.93% |
| Yan | Unrestricted (males) | PCR | 11.9 | 8.4–16.9 | | 87.83% |
| Yan | Unrestricted (adults) | PCR | 23.2 | 19.0–28.4 | | 99.24% |
| Yan | Unrestricted (children) | PCR | 9.9 | 8.1–12.2 | | 85.74% |
| Yan | Unrestricted (with chronic diseases) | PCR | 24.2 | 19.2–30.2 | | 84.07% |

*Appendix 1—table 13 Continued on next page*

*Appendix 1—table 13 Continued*

| Study | Sampling site (subgroups) | Laboratory method | Pooled estimates (mean/median) | 95% CI | Range | I² |
|---|---|---|---|---|---|---|
| Yan | Unrestricted (without chronic diseases) | PCR | 11.5 | 5.3–25.0 | | 82.11% |
| Yan | Unrestricted (treated with corticosteroid) | PCR | 28.3 | 25.6–31.2 | | 0.00% |
| Yan | Unrestricted (treated without corticosteroid) | PCR | 16.2 | 11.5–22.5 | | 92.27% |
| Yan | Unrestricted (antiviral treatment) | PCR | 17.6 | 13.4–22.2 | | 98.99% |
| Yan | Unrestricted (mono-antiviral treatment) | PCR | 21.2 | 15.3–29.2 | | 90.04% |
| Yan | Unrestricted (multiantiviral treatment) | PCR | 20.3 | 13.7–30.3 | | 99.46% |
| Chen | Unrestricted (asymptomatic infections) | PCR or serum antibody | 14.14* | 11.25–17.04 | 11.00–17.25 | |

*Median estimate, **median/grouped median, ***this study analyzed by cases, and reported mean ± SD.

**Appendix 1—table 14.** Pooled estimates for duration of infectious period for COVID-19 in subgroups from two systematic reviews.

| Study | Sampling site (subgroups) | Laboratory method | Pooled estimates | 95% CI | Range | I² |
|---|---|---|---|---|---|---|
| Rahmani | Replicant competent virus isolation | Viral culture | 7.27 | 5.70–8.84 | 3.40–89.00 | 92.2% |
| Wang | Not specific | Not specific | 6.25 | 5.09–7.51 | | |
| Rahmani | Replicant competent virus isolation (immunocompetent individuals) | Viral culture | 6.33 | 4.92–7.75 | 3.00–13.00 | 92.4% |
| Rahmani | Replicant competent virus isolation (immunocompromised individuals) | Viral culture | 29.50 | 12.46–46.53 | 13.80–89.00 | 84.8% |

**Appendix 1—table 15.** Probability distributions of the incubation period and relative infectivity levels during the infectious period.

For the infectious period, day 0 corresponds to the symptom onset day. These two distributions are used to generate the infectiousness profile since infections.

| | Incubation period | | Infectious period |
|---|---|---|---|
| Day | Mean = 5 days | Day | Max = 13 days |
| 1 | 0.058 | −5 | 1.0 |
| 2 | 0.11 | −4 | 1.0 |
| 3 | 0.14 | −3 | 1.0 |
| 4 | 0.16 | −2 | 1.0 |
| 5 | 0.15 | −1 | 1.0 |
| 6 | 0.13 | 0 | 1.0 |
| 7 | 0.10 | 1 | 1.0 |
| 8 | 0.068 | 2 | 1.0 |
| 9 | 0.044 | 3 | 0.8 |
| 10 | 0.026 | 4 | 0.6 |
| 11 | 0.014 | 5 | 0.4 |
| 12 | 0.0072 | 6 | 0.2 |
| 13 | 0.0034 | 7 | 0.1 |
| 14 | 0.0015 | 8 | |

