## [Editor Report]

While it has been demonstrated that for SARS-CoV-2, a small fraction of individuals contributes to the majority of onward transmission, this heterogeneity is driven by multiple factors that span both biological and behavioral causes. By performing a solid meta-analysis of household transmission studies, the authors fit a household transmission model to the curated data to estimate variation in infectiousness which provides a valuable contribution to the existing knowledge base. By collating data from multiple studies, they are able to more fully investigate individual variability.

---

## [Decision Letter]

**Decision letter after peer review:**

Thank you for submitting your article "The effect of variation of individual infectiousness on SARS-CoV-2 transmission in households" for consideration by *eLife*. Your article has been reviewed by 2 peer reviewers, and the evaluation has been overseen by a Reviewing Editor and Miles Davenport as the Senior Editor. The reviewers have opted to remain anonymous.

Essential revisions:

1) Both reviewers have highlighted a number of areas where additional clarification (particularly for σ_var) which currently does not have sufficient detail/interpretation to warrant publication.

2) Questions about parameter identifiability and model validation were not sufficiently addressed – particularly in light of the many individual-level parameters estimated in the analysis.

3) Additional exploration of empirical relationships (such as individual infectiousness and household size, as an example) should be explored. For example, how much of the heterogeneity is driven by biological factors (such as being the primary case).

4) Additional work should be done to help disentangle differences between biological or behavioral factors since this will greatly change the interpretation of inferred parameters. In addition, more contextual information should be provided (socio-economic status and the implications of this as a confounder, a more detailed exploration of interventions that were in place and their implications) should be done.

*Reviewer #1 (Recommendations for the authors):*

Tsang et al. evaluate the degree of transmission heterogeneity among COVID-19 cases using data compiled from 17 household transmission studies. Transmission heterogeneity and super spreading are important but incompletely understood drivers of epidemic dynamics. While it is clear that substantial heterogeneity exists among infectors, but the relative contribution of behavior (e.g. the number of contacts), timing (e.g. large number of social interactions near the peak of infectiousness), and intrinsic biological differences in infectiousness are not resolved.

Tsang et al. compiled household transmission data from 17 published studies. They independently fit a model to each data set in order to quantify transmission heterogeneity arising from intrinsic biological differences in infectiousness, after controlling for differences in the number of contacts (i.e. household size), and timing (via the inferred time since infection). Repeated application of the same model to multiple datasets is a strength of this study because it allows the authors to assess associations between the inferred parameters and aspects of study design that can influence case ascertainment.

The authors estimate substantial intrinsic transmission heterogeneity via the parameter σ_var, which could be attributed to individual differences in viral loads or differences in the infector's contact intensity with others in the household. They show that intrinsic, within-household variation is associated with other population-level metrics of transmission heterogeneity, including p_80 and p_0.

Although the study focuses on the household size as a confounder of heterogeneity in infectiousness, the ways that the model controls for household size and the interpretation of the inferred heterogeneity parameter, σ_var could be explained more clearly. Furthermore, it seems that basic empirical relationships between individual infectiousness (δ_i) and household size are not explored. Finally, questions about parameter identifiability and model validation could be addressed more extensively, especially in light of the many individual-level parameters estimated in the analysis.

Overall, the study design is sound and the analysis of multiple datasets is a strength. However, the modeling approach and the biological interpretation/significance of the results could be more clearly explained. Also, given the large number of individual-level parameters estimated, it would be ideal to see a direct assessment of parameter identifiability (e.g. correlation plots), and model validation on simulated data (i.e. can the model estimate known parameters from simulated data?). The latter shouldn't be too difficult, given that the authors already have a simulation model for tests of model adequacy.

Specific comments:

1. Please explain assumptions about the individual infectiousness profile f(.) (from section 2 of the Materials and methods). What is the assumed shape of this function? How is it parameterized? How could the inclusion of this function in the model interact with inferred individual times of infection, especially if times of infection are inaccurate? On a related note, the authors might consider introducing the concept of contact timing as a factor that can influence heterogeneity in transmission.

2. The household size is a key conceptual focus of the study, but the main text methods/results don't explain clearly how the model accounts for household size or contact number. My interpretation is that the model accounts for household size in two ways. First. the model estimates the per-contact hazard of infection, which implicitly controls for household size/contact number. Second, the model includes a parameter, β, which represents a dilution effect, wherein the transmission hazard might be diluted in larger households due to less per-individual contact intensity. I'm not sure that the dilution effect is currently explained anywhere in the main text, and it could also be explained more clearly that the model is designed to estimate transmission heterogeneity after adjusting for household size.

2a. It would be helpful if the authors could discuss the distinction between contact number, contact frequency, and contact duration somewhere in the text.

3. Given the focus on household size and the number of contacts, it seems there are missed opportunities to explore how the inferred level of individual infectiousness (δ_i) co-varies with metrics like household size.

4. Please explain in more detail the meta-regression sensitivity analyses.

5. I'm not sure that tables S6-S9 are cross-referenced anywhere in the main text, and their legends aren't sufficiently detailed to fully communicate how to interpret these tables in the context of the broader study.

*Reviewer #2 (Recommendations for the authors):*

Transmission heterogeneity and superspreading of SARS-CoV-2 have been demonstrated repeatedly through real-world observational studies, with a small fraction of individuals accounting for the majority of onward transmission. However, this observed transmission heterogeneity is likely a superposition of accumulated variations from multiple factors, including but not limited to host behavioral factors such as the variation in contact numbers contact duration, and contact settings; variations in the adoption of preventive measures (mask-wearing, physical distancing, etc.); the effects of NPIs (case isolations and contact quarantines, population-level lockdowns); as well as biological factors including differences in shedding duration and intensity of the primary case, variation in susceptibility among close contacts. Fewer studies, however, attempted to isolate the effects of individual factors contributing to the overall transmission heterogeneity, while controlling for other factors. In the manuscript entitled "The effect of variation of individual infectiousness on SARS-CoV-2 transmission in households", the authors aim at characterizing the variation in individual infectiousness of SARS-CoV-2, controlling for other host factors. To achieve this, the authors performed a meta-analysis of household transmission studies conducted during or not too longer after the initial SARS-CoV-2 wave caused by the ancestral strain across the globe. The authors fitted a household transmission model to the curated data and estimated the variation in infectiousness by introducing random effects of individual infectiousness and its population-level distribution.

The study has several strengths. First, by choosing analyzing data from the household study, the authors were able to control for several key factors contributing to the overall transmission heterogeneity but not variation in individual infectiousness, including the number of contacts as well the setting of transmission (household). The authors also incorporated an additional parameter in the household transmission model to adjust for the impact of household size on household transmission risk, which has been identified as an important risk factor for SARS-CoV-2 transmission within the household across multiple studies. The authors also curated the studies during the early stage of the pandemic so that most household contacts would remain naïve during the study period, and the immune status of the household contact (either due to prior infection or vaccination) would be unlikely to confound the results of the study.

However, the study also has a few limitations. First, it is difficult to disentangle if the observed variation in infectiousness is due to biological factors or behavioral factors. During the study period of interest, public health agencies across the globe were recommending at-home isolation guidelines aiming at reducing transmission within the household, including mask-wearing when in contact with other household members, using separate bedrooms/bathrooms, avoiding having meals together, etc. The differences in the guidelines across nationals/regions as well as the level of compliance with guidelines at the household level would also impact individual household transmission risk. Second, the risk of acquiring infections from the community could be heavily influenced by the socio-economic status, since multiple studies have clearly demonstrated stark disparity of COVID-19 burden, factors such as occupation (essential workers tend to be low-wage jobs) assess to PPE and healthcare were likely to contribute to the observed disparity. Lastly, it is also difficult to entangle if the observed heterogeneity is due to the biological factors of the primary case (i.e., variation in shedding duration/intensity) or the contact (variation in susceptibility). The current formulation of the transmission model only addresses the former not the latter.

For the household transmission model concerning imputation of the timing of infection, it is unclear if the timing of infection is partially missing, or if the timing of infection were unavailable for all studies. If it is the former case, the authors should give a more detailed description of the imputation process in a study-by-study fashion and report the proportion of infection time that was imputed. If it is the latter case, I do not understand the benefit of explicitly modeling the temporal infectiousness profile, thus I would recommend the authors use a simpler chain-binomial model fitting to only the binary outcome (of infected or not) unless the authors make a convincing case otherwise.

Both p80 and p0 are poor statistics to characterize household transmission due to the discrete nature of household contact numbers. For example, it is unintuitive to interpret and compare p80 for a household size of 2 vs 10 (for a household size of 2 conditional on an index case within the household this could only be interpreted as the proportion of households with secondary infections, which is cross-household level statistics, while for a household size of 10, it can be interpreted as a meaningful statistical for each household). Furthermore, the interpretation of p80 as a metric for transmission heterogeneity becomes even murkier if multi-generation transmission within the household is considered, especially for a larger household. For p0, this metric is heavily dependent on household size. For example, one would expect the p0 for a household size of 10 (with on index) to be way smaller than p0 household size of 2 (assuming no dilution effects on transmission with the increase of household size), as in the former case p0 characterize the probability of all 9 uninfected contacts escaping infection from the index, while in the latter p0 characterizes the probability to escape infection by only 1 household contact. Thus, it is meaningless to compare p0 without controlling for household size across studies. For the arguments above, I recommend the authors remove both p80 and p0 from the paper.

Through the formulation of the household transmission model, the authors essentially assume a lognormal distribution of individual infectiousness, which is a long-tailed distribution by nature. I think this is an important point that needs to be highlighted. Has the author tried a normal distribution instead of a log-normal distribution? Would a normal distribution fit better or worse to the data than the log-normal one? I recommend the authors conduct a sensitivity analysis of a normal distribution of the individual infectiousness (i.e., not taking the exponential of the σ term for the hazard of infection function on page 7).

Table S5 is cropped.

---

## [Author Response]

Essential revisions:1) Both reviewers have highlighted a number of areas where additional clarification (particularly for σ_var) which currently does not have sufficient detail/interpretation to warrant publication.

We have modified the discussion to talk about the interpretation and limitation of infectiousness variation. We clarified that the estimated infectiousness variations measures the variation caused by difference in individuals, which may be contributed by both biological factors and host behaviors, but not other environmental factors. Please see Response 7 for more details.

We also added a limitation that our approach could not disentangle the observed heterogeneity from biological factors affecting infectiousness and variations in susceptibility of contacts. Please see Response 14 for more details.

2) Questions about parameter identifiability and model validation were not sufficiently addressed – particularly in light of the many individual-level parameters estimated in the analysis.

We have added the correlation plots for the posterior distribution of parameters, and also rerun a longer MCMC chain (with 500K iterations instead of 100K iterations) to ensure that our estimates of infectiousness variations are valid. Please see Response 7 for more details.

3) Additional exploration of empirical relationships (such as individual infectiousness and household size, as an example) should be explored. For example, how much of the heterogeneity is driven by biological factors (such as being the primary case).

The individual infectiousness parameters are nuisance parameter in our work, since there is no individual level information on the case and contacts. However, our model is designed in a way that covariates affecting susceptibility or infectiousness could be added if individual level information is available. Please see Response 11 for more details.

We also conduct meta-analysis and meta-regression to determine if any factors may be associated with the infectiousness variation. We found that this is not associated with the mean numbers of contacts and household size, and only associated with if the study used PCR to confirm secondary cases only (Table S11).

4) Additional work should be done to help disentangle differences between biological or behavioral factors since this will greatly change the interpretation of inferred parameters. In addition, more contextual information should be provided (socio-economic status and the implications of this as a confounder, a more detailed exploration of interventions that were in place and their implications) should be done.

We have extended our discussion on difference between biological and behavioral factors. In summary, behavioral factors including difference in frequency and duration of contacts caused by different factors, like the role in family, recommendation from public health agencies, among individuals. Biological factors include difference in viral shedding and infectious period among individuals. We also provide more information on other confounding factors, such as social-economic status, recommendation from public health agencies, the implementation of lockdowns that may increase the time spending at home. These may increase of decrease the SAR, which is shown to be negatively correlated with infectiousness variations. Please see Response 7 and 14 for more details.

Reviewer #1 (Recommendations for the authors):Tsang et al. evaluate the degree of transmission heterogeneity among COVID-19 cases using data compiled from 17 household transmission studies. Transmission heterogeneity and super spreading are important but incompletely understood drivers of epidemic dynamics. While it is clear that substantial heterogeneity exists among infectors, but the relative contribution of behavior (e.g. the number of contacts), timing (e.g. large number of social interactions near the peak of infectiousness), and intrinsic biological differences in infectiousness are not resolved.Tsang et al. compiled household transmission data from 17 published studies. They independently fit a model to each data set in order to quantify transmission heterogeneity arising from intrinsic biological differences in infectiousness, after controlling for differences in the number of contacts (i.e. household size), and timing (via the inferred time since infection). Repeated application of the same model to multiple datasets is a strength of this study because it allows the authors to assess associations between the inferred parameters and aspects of study design that can influence case ascertainment.The authors estimate substantial intrinsic transmission heterogeneity via the parameter σ_var, which could be attributed to individual differences in viral loads or differences in the infector's contact intensity with others in the household. They show that intrinsic, within-household variation is associated with other population-level metrics of transmission heterogeneity, including p_80 and p_0.Although the study focuses on the household size as a confounder of heterogeneity in infectiousness, the ways that the model controls for household size and the interpretation of the inferred heterogeneity parameter, σ_var could be explained more clearly. Furthermore, it seems that basic empirical relationships between individual infectiousness (δ_i) and household size are not explored. Finally, questions about parameter identifiability and model validation could be addressed more extensively, especially in light of the many individual-level parameters estimated in the analysis.Overall, the study design is sound and the analysis of multiple datasets is a strength. However, the modeling approach and the biological interpretation/significance of the results could be more clearly explained. Also, given the large number of individual-level parameters estimated, it would be ideal to see a direct assessment of parameter identifiability (e.g. correlation plots), and model validation on simulated data (i.e. can the model estimate known parameters from simulated data?). The latter shouldn't be too difficult, given that the authors already have a simulation model for tests of model adequacy.

Thank you for your suggestions. In brief, we summarized that the estimated infectiousness variations measures the variation caused by difference in individuals, which may be affected by both biological factors and host behaviors but not other confounding factors, such as the difference in number of contacts, pre-existing immunity among contacts or difference in transmissibility among variants. We also added more details on the interpretation of the estimated infectiousness variation.

We added ‘Given that we focused our analysis on households with known number of contacts in studies conducted in the early outbreaks caused by ancestral strains, the estimated infectiousness variations are corrected for the variations caused by number of contacts and transmission risks in different settings, difference in pre-existing immunity among contacts (almost everyone is naïve and unvaccinated), and the difference in transmissibility among variants. Hence, this estimated infectiousness variation measures the variation caused by difference in individuals, which may be contributed by both biological factors and host behaviors, but not other confounding factors such as…. ‘ in the Discussion section in the revised manuscript.

We conducted a simulation study, and found there is no important systematic bias, and that 88-100% of 95% credible interval covered the simulation value. We also added the correlation plots in the revised manuscript. The correlation between parameters are moderate. We run long MCMC chain to ensure our results are valid. We conduct a sensitivity by running the MCMC chain with 500,000 iterations and the results are similar to the one with 100,000 iterations. This suggests that 100,000 iterations are enough (our main results).

We added ‘Simulation studies demonstrated there is no important systematic bias, with 88 to 100% of the 95% credible intervals covering the simulation value (Appendix 1 – Table 8). This suggests that the algorithm could estimate adequately the posterior distribution.’ In the Result section of the revised manuscript.

Specific comments:1. Please explain assumptions about the individual infectiousness profile f(.) (from section 2 of the Materials and methods). What is the assumed shape of this function? How is it parameterized? How could the inclusion of this function in the model interact with inferred individual times of infection, especially if times of infection are inaccurate? On a related note, the authors might consider introducing the concept of contact timing as a factor that can influence heterogeneity in transmission.

We apologize for the confusion. We added a table to summarize the assumed values extracted from the reference papers, and a plot for the assumed individual infectiousness profile function f(.). We also moved the exact formula and more details of the model from appendix to the main text, to enhance the readability.

In the estimation, the infectiousness profile are used to determine the generation time, and it would influence the observed heterogeneity of distribution of number of secondary cases (including all generations). When there is no difference in infectiousness variation among cases, if the generation time is shorter, there will be more generations of cases, and hence the observed distribution of number of secondary cases would be more heterogeneous. Hence, this assumption is setting a ‘baseline’ of observed heterogeneity of observed distribution of number of secondary cases when there is no difference in infectiousness among cases, so that the model could estimate the extra heterogeneity due to variations of individual infectiousness.

We thank for the reviewer’s suggestion about contact timing, which is a good point and we also added in the Discussion. We added ‘the duration of contact could vary, which could also contribute to the variations in infectiousness of cases.’ in the Discussion section of the revised manuscript.

2. The household size is a key conceptual focus of the study, but the main text methods/results don't explain clearly how the model accounts for household size or contact number. My interpretation is that the model accounts for household size in two ways. First. the model estimates the per-contact hazard of infection, which implicitly controls for household size/contact number. Second, the model includes a parameter, β, which represents a dilution effect, wherein the transmission hazard might be diluted in larger households due to less per-individual contact intensity. I'm not sure that the dilution effect is currently explained anywhere in the main text, and it could also be explained more clearly that the model is designed to estimate transmission heterogeneity after adjusting for household size.

We thank the reviewer’s a comprehensive summary on the way that the model accounts for the household size or number of household contacts. We added this information, including the dilution effect, to the revised manuscript as follows:

We added ‘the model could estimate the per-contact hazard of infection, which implicitly controls for number of household contacts in households.’ In the Result section of revised manuscript.

We added ‘In the model, the hazard of infection of individual *j* at time *t* from an infected household member *i,* with infection time *t_i_* in household *k*, was λi→j(t)=λhXkβ ∗exp⁡(δi)∗f(t−ti) where λh was the baseline hazard, δi followed a Normal distribution with mean 0 and standard deviation σvar, which quantified the variation of individual infectiousness (hereafter denoted as infectiousness variation). The relative infectiousness of case *i* compared with case *j* was exp(δi)/exp(δj).

Xk was the number of household contacts. β was the parameter describing the relationship between number of household contacts and transmission rate. It ranged from 0 to 1, with 0 indicating that the transmission rate was independent of number of household contacts while 1 indicated that the transmission rate was inversely proportional to number of household contacts (i.e. dilution effect of the contact time per contact which was lower when the number of household contacts is higher).’ to the Method section of the revised manuscript.

2a. It would be helpful if the authors could discuss the distinction between contact number, contact frequency, and contact duration somewhere in the text.

Thank you for your suggestion. We agree this is an important point and added to the revised manuscript. We added ‘It should be noted that there are several components in the contacts, including the numbers, frequencies and duration of contacts that may contribute to the infectiousness variations. Previous studies also suggest that their relative importance in transmission may be different among viruses.’ In the Discussion in the revised manuscript.

3. Given the focus on household size and the number of contacts, it seems there are missed opportunities to explore how the inferred level of individual infectiousness (δ_i) co-varies with metrics like household size.

Thank you for your suggestion. We estimated that the posterior correlation between inferred level of individual infectiousness (δ_i) and household size are not significant ( the credible intervals are all covered one). This suggested the inferred individual infectiousness was not correlated with household size.

**Author response table 1. sa2table1:** 

Study	Pearson’s correlation	Shearman’s correlation
Lyngse, et al.	0.00 (-0.02, 0.03)	0.00 (-0.03, 0.02)
Carazo, et al.	0.00 (-0.03, 0.04)	0.00 (-0.03, 0.03)
Laxminarayan, et al.	0.00 (-0.07, 0.05)	-0.02 (-0.08, 0.03)
Dattner, et al.	0.01 (-0.07, 0.09)	0.00 (-0.08, 0.09)

4. Please explain in more detail the meta-regression sensitivity analyses.

Sorry for the confusion. Meta-analysis and Meta-regression is conducted to pool the estimates of infectiousness variation, which was estimated separately for 14 identified studies. Furthermore, it allows us to determine the potential association between estimated infectiousness variation and study characteristics, including the following factors: the mean and SD of number of household contacts, implementation of lockdown, ascertainment method of index and secondary cases, only ancestral strains are circulating in study period, and all household contacts were tested. We have clarified in the Method section of the revised manuscript.

5. I'm not sure that tables S6-S9 are cross-referenced anywhere in the main text, and their legends aren't sufficiently detailed to fully communicate how to interpret these tables in the context of the broader study.

These tables summarize the systematic review and meta-analysis about viral shedding of COVID-19 cases, which is a proxy of individual infectiousness of cases. They showed heterogeneity in the level of viral shedding of COVID-19 cases, which support our results about variation of individual infectiousness.

Reviewer #2 (Recommendations for the authors):Transmission heterogeneity and superspreading of SARS-CoV-2 have been demonstrated repeatedly through real-world observational studies, with a small fraction of individuals accounting for the majority of onward transmission. However, this observed transmission heterogeneity is likely a superposition of accumulated variations from multiple factors, including but not limited to host behavioral factors such as the variation in contact numbers contact duration, and contact settings; variations in the adoption of preventive measures (mask-wearing, physical distancing, etc.); the effects of NPIs (case isolations and contact quarantines, population-level lockdowns); as well as biological factors including differences in shedding duration and intensity of the primary case, variation in susceptibility among close contacts. Fewer studies, however, attempted to isolate the effects of individual factors contributing to the overall transmission heterogeneity, while controlling for other factors. In the manuscript entitled "The effect of variation of individual infectiousness on SARS-CoV-2 transmission in households", the authors aim at characterizing the variation in individual infectiousness of SARS-CoV-2, controlling for other host factors. To achieve this, the authors performed a meta-analysis of household transmission studies conducted during or not too longer after the initial SARS-CoV-2 wave caused by the ancestral strain across the globe. The authors fitted a household transmission model to the curated data and estimated the variation in infectiousness by introducing random effects of individual infectiousness and its population-level distribution.The study has several strengths. First, by choosing analyzing data from the household study, the authors were able to control for several key factors contributing to the overall transmission heterogeneity but not variation in individual infectiousness, including the number of contacts as well the setting of transmission (household). The authors also incorporated an additional parameter in the household transmission model to adjust for the impact of household size on household transmission risk, which has been identified as an important risk factor for SARS-CoV-2 transmission within the household across multiple studies. The authors also curated the studies during the early stage of the pandemic so that most household contacts would remain naïve during the study period, and the immune status of the household contact (either due to prior infection or vaccination) would be unlikely to confound the results of the study.However, the study also has a few limitations. First, it is difficult to disentangle if the observed variation in infectiousness is due to biological factors or behavioral factors. During the study period of interest, public health agencies across the globe were recommending at-home isolation guidelines aiming at reducing transmission within the household, including mask-wearing when in contact with other household members, using separate bedrooms/bathrooms, avoiding having meals together, etc. The differences in the guidelines across nationals/regions as well as the level of compliance with guidelines at the household level would also impact individual household transmission risk. Second, the risk of acquiring infections from the community could be heavily influenced by the socio-economic status, since multiple studies have clearly demonstrated stark disparity of COVID-19 burden, factors such as occupation (essential workers tend to be low-wage jobs) assess to PPE and healthcare were likely to contribute to the observed disparity. Lastly, it is also difficult to entangle if the observed heterogeneity is due to the biological factors of the primary case (i.e., variation in shedding duration/intensity) or the contact (variation in susceptibility). The current formulation of the transmission model only addresses the former not the latter.

We appreciate the reviewer’s very comprehensive summary of our work and unique insight of the interpretation of our results, and we added many of them in the revised manuscript. We expanded the section of our discussion listing the different limitations, in particular the fact that our approach can not disentangle the observed heterogeneity from biological factors affecting infectiousness and variations in susceptibility of contacts.

We added the followings in the Discussion in the revised manuscript:

“Given that we focused our analysis on households with known number of contacts in studies conducted in the early outbreaks caused by ancestral strains, the estimated infectiousness variations do not include the variations caused by number of contacts and transmission risks in different settings, difference in pre-existing immunity among contacts (almost everyone is naïve and unvaccinated), and the difference in transmissibility among variants. Hence, this estimated infectiousness variation measures the variation caused by difference in individuals, which may be contributed by both biological factors and host behaviors, but not other confounding factors.”

“It should be noted that there are several components in the contacts, including the numbers, frequencies and duration of contacts that may contribute to the infectiousness variations. Previous studies also suggest that their relative importance in transmission may be different among viruses. Individual behaviors may also be influenced by the implemented control measures and recommendation from public health agencies. For example, lockdown and stay-at-home orders may increase the time spending at home. Mask-wearing when in contact with other household members, using separate bedrooms and bathrooms, avoiding having meals together may reduce the transmission in households. Also, social disparity such as occupation may increase or reduce the risk of transmission in households, including the availability of personal protective equipment (PPE), or being healthcare worker. These may have impacted the SAR in households, but also estimated infectiousness variation in this setting.”

“In addition, we could not include factors affecting susceptibility to infection. Our estimates of infectiousness variation should be interpreted in light of these limitations: they capture heterogeneity in infectiousness due to demographic, host and biological factors. However, in one study that included susceptibility component in the estimation of individual infectiousness, substantial heterogeneity remained with 20% of cases estimated to contribute to 80% of transmission.”

“Third, we assumed that risks of infection from community for all households are the same, but there were different factors that may affecting this, including occupations, such as healthcare workers, social economic status that related to assess to personal protective equipment”

For the household transmission model concerning imputation of the timing of infection, it is unclear if the timing of infection is partially missing, or if the timing of infection were unavailable for all studies. If it is the former case, the authors should give a more detailed description of the imputation process in a study-by-study fashion and report the proportion of infection time that was imputed. If it is the latter case, I do not understand the benefit of explicitly modeling the temporal infectiousness profile, thus I would recommend the authors use a simpler chain-binomial model fitting to only the binary outcome (of infected or not) unless the authors make a convincing case otherwise.

Sorry of confusion. The timing of infection were unavailable for all studies. The reason of not using a simple chain-binomial model is that it is impossible to add the parameters characterizing individual infectiousness (δ_i) in the model. When there are different types of susceptible in data, chain-binomial model required adding separate probability parameters for each type of susceptible in the model, with each type of transmission (from household members or from outside households).

We have clarified this by adding ‘Since the data were extracted from publication, the infection time of all cases was unavailable for all studies.’ in the Method section in the revised manuscript. We also added a section in Appendix to explain this.

Both p80 and p0 are poor statistics to characterize household transmission due to the discrete nature of household contact numbers. For example, it is unintuitive to interpret and compare p80 for a household size of 2 vs 10 (for a household size of 2 conditional on an index case within the household this could only be interpreted as the proportion of households with secondary infections, which is cross-household level statistics, while for a household size of 10, it can be interpreted as a meaningful statistical for each household). Furthermore, the interpretation of p80 as a metric for transmission heterogeneity becomes even murkier if multi-generation transmission within the household is considered, especially for a larger household. For p0, this metric is heavily dependent on household size. For example, one would expect the p0 for a household size of 10 (with on index) to be way smaller than p0 household size of 2 (assuming no dilution effects on transmission with the increase of household size), as in the former case p0 characterize the probability of all 9 uninfected contacts escaping infection from the index, while in the latter p0 characterizes the probability to escape infection by only 1 household contact. Thus, it is meaningless to compare p0 without controlling for household size across studies. For the arguments above, I recommend the authors remove both p80 and p0 from the paper.

Thank you. We agree with the reviewer and removed p80 and p0 from the paper.

Through the formulation of the household transmission model, the authors essentially assume a lognormal distribution of individual infectiousness, which is a long-tailed distribution by nature. I think this is an important point that needs to be highlighted. Has the author tried a normal distribution instead of a log-normal distribution? Would a normal distribution fit better or worse to the data than the log-normal one? I recommend the authors conduct a sensitivity analysis of a normal distribution of the individual infectiousness (i.e., not taking the exponential of the σ term for the hazard of infection function on page 7).

Thank you for this suggestion. We conducted a sensitivity analysis for assuming the individual infectiousness followed a Γ distribution, which has a shorter tail compared with lognormal distribution. However, model comparison suggested that assuming γ distribution for individual infectiousness parameter performed substantially worse, compared with the lognormal distribution in the main analysis (Appendix 1 – Table 9).

Table S5 is cropped.

Sorry for the mistake, we revised the table.